# FM-IRL: Flow-Matching for Reward Modeling and Policy Regularization in Reinforcement Learning

## Abstract

Flow Matching (FM) has shown remarkable ability in modeling complex distributions and achieves strong performance in offline imitation learning for cloning expert behaviors. However, despite its behavioral cloning expressiveness, FM-based policies are inherently limited by their lack of environmental interaction and exploration. This leads to poor generalization in unseen scenarios beyond the expert demonstrations, underscoring the necessity of online interaction with environment. Unfortunately, optimizing FM policies via online interaction is challenging and inefficient due to instability in gradient computation and high inference costs. To address these issues, we propose to let a "student" policy with simple MLP structure to explore the environment and be online updated via RL algorithm with a reward model. This reward model is associated with a "teacher" FM model, containing rich information of expert data distribution. Furthermore, the same "teacher" FM model is utilized to regularize the "student" policy's behavior to stabilize policy learning. Due to the student's simple architecture, we avoid the gradient instability of FM policies and enable efficient online exploration, while still leveraging the expressiveness of the teacher FM model. Extensive experiments shows that our approach significantly enhances learning efficiency, generalization, and robustness, especially when learning from suboptimal expert data.

## 1 Introduction

Flow Matching (FM) has emerged as a powerful generative modeling approach, capable of efficiently fitting multi-modal data distributions. Compared to diffusion models, FM offers faster sampling and more stable training, making FM particularly well-suited for applications that require both high generation quality and computational efficiency (Liu et al., 2025; Lipman et al., 2022).

Building on its strong distribution-matching capabilities and efficiency, FM has been extended to imitation learning (IL) in robot locomotion, manipulation and navigation (Zhang et al., 2025b). For example, flow matching policy (Wang et al., 2022; Braun et al., 2024; Zhang et al., 2025a) is utilized for capturing complex, multi-modal action distribution from expert demonstrations, enabling the agent to achieve good performance via cloning expert behavior. However, a key limitation arises: FM policies are inherently designed for offline learning (following behavior cloning (BC) paradigm) and lack a built-in exploration mechanism (Zhang et al., 2025b). Accordingly, FM policies struggle to generalize to new scenarios not covered in the expert demonstrations. Furthermore, when the expert data is limited or sub-optimal, BC-based FM policies will overfit to sub-optimal data and the lack of exploration can lead to poor generalization and performance degradation (Yu et al., 2025; Wan et al., 2025b; 2024), as shown in experiment at Section 4.3. To address this limitation, a natural potential direction is to apply FM policies beyond BC paradigm and let it be updated during the online exploration with the environment.

Despite the aforementioned limitations of BC-based FM policies, several works have successfully extended diffusion policies beyond the BC framework. For example, DQL (Wang et al., 2022) was proposed to integrate diffusion policy with Q-learning, but still limited to offline RL setting. More importantly, when the diffusion policies or FM policies need to be optimized via online policy gradient methods (such as Proximal Policy Optimization (Schulman et al., 2017)), it becomes

challenging to compute accurate gradients due to the iterative nature of diffusion or FM policies. Specifically, the policy gradients need to be computed via stochastic simulation techniques, such as re-parametrization. This arises the demand of backpropagation through time due to the recursion in numerical ODE solvers (e.g., the Euler method), which is extremely unstable and costly in practice (Park et al., 2025) (for the technical details, please refer to Appendix D).

On the other hand, inverse reinforcement learning (IRL) supports online policy update when we are not accessible to the reward function of environment but are given the expert demonstrations, since IRL learns a reward function from expert demonstrations and uses the learned reward function to guide the online environmental exploration and policy optimization (Ng et al., 2000). Therefore, a natural question arises: *can we utilize IRL to empower FM policies with online exploratory strength, while bypassing the unstableness of FM policies' gradient computation?*

Inspired by these insights, we propose a novel framework termed Flow Matching Inverse Reinforcement Learning (FM-IRL), which simultaneously addresses three key limitations of conventional FM policies: (1) lack of exploration with environment, (2) challenges in computing accurate policy gradients for online optimization, and (3) high computational cost during inference. To enhance FM policies with online exploration and address the associated challenges, our approach employs a teacher-student architecture, where an FM model acts as the teacher to guide a lightweight, MLP-based student policy (also behavioral policy) that interacts directly with the environment. The FM teacher fulfills two roles: (1) an FM-enhanced reward model that turns distribution-level FM distances into a shaped reward for online RL, and (2) an FM-based regularizer that prevents the agent's behavior from deviating significantly from the expert distribution to avoid poor reward estimation and unstable training of reward model. Through this design, the student policy can actively interact with the environment while enjoying stable and efficient gradient updates, since only the MLP parameters are optimized. At deployment, the student policy alone is used, yielding fast inference suitable for real-world applications. Crucially, the rich distributional knowledge of the FM teacher is effectively "infused" into the student policy via the reward model and regularizer, thereby leveraging the strong distribution-matching capacity of FM while circumventing its inherent limitations.

To our best knowledge, our work is the first one to utilize FM for reward modeling and regularization in online reinforcement learning. We justified the superiority of FM-IRL via the learning efficiency study, generalization study, study of robustness to sub-optimal expert data, ablation study, and a case study in Section 4. We summarized our contribution as follows:

- We propose FM-IRL, the first framework that seamlessly utilize IRL to empower FM policy with exploratory strength.
- We adopts a teacher-student architecture using an FM-enhanced discriminator and regularizer to bypass the key challenge of applying FM policies in online environment, and improve the inference efficiency during deployment.
- Compared to FM policy, the empirical results show that our method significantly improves the generalization of learned policy and the robustness against sub-optimal expert data.

## 2 BACKGROUND

### 2.1 FLOW MATCHING (FM) FOR GENERATIVE MODELING

Following the generative modeling paradigm, FM aims to learn a generator $G_\theta$ that produces samples aligning with a target distribution, based on samples from this distribution. FM assumes the existence of a continuous flow that transforms a simple initial distribution $\mathcal{N}(0, I)$ at $t = 0$ to the target distribution at $t = 1$. This transformation is defined by the ordinary differential equation:

$$\frac{d}{dt}\phi_t(x) = v_t(\phi_t(x)), \quad \phi_0(x) = x.$$

The core idea is to regress a parametric vector field $v_t(x; \theta)$ toward a target vector field $u_t(x)$ that generates a desired probability density path $p_t(x)$. This is formalized through the FM objective:

$$\mathcal{L}_{\text{FM}}(\theta) = \mathbb{E}_{t, p_t(x)} \|v_t(x) - u_t(x)\|^2,$$

where $t \sim \mathcal{U}[0, 1]$ and $x \sim p_t(x)$. Since the marginal vector field $u_t(x)$ and probability path $p_t(x)$ are generally intractable, Conditional Flow Matching (CFM) (Lipman et al., 2022) provides

a practical alternative. Let $q(x_1)$ be the data distribution and $p_t(x|x_1)$ be a conditional probability path satisfying $p_0(x|x_1) = p(x)$ and $p_1(x|x_1) = \mathcal{N}(x_1, \sigma_{\min}^2 I)$. The marginal path is:

$$p_t(x) = \int p_t(x|x_1)q(x_1)dx_1 = \mathbb{E}_{x_1}[p_t(x|x_1)].$$

The CFM objective is defined as:

$$\mathcal{L}_{\text{CFM}}(\theta) = \mathbb{E}_{t,q(x_1),p_t(x|x_1)} \left\| v_t(x) - u_t(x|x_1) \right\|^2.$$

Crucially, it is shown that $\nabla_\theta \mathcal{L}_{\text{FM}}(\theta) = \nabla_\theta \mathcal{L}_{\text{CFM}}(\theta)$, making CFM a tractable training objective.

A common choice for the conditional path is a Gaussian parameterization: $p_t(x|x_1) = \mathcal{N}\left(x \mid \mu_t(x_1), \sigma_t(x_1)^2 I\right)$, with boundary conditions $\mu_0(x_1) = 0$, $\sigma_0(x_1) = 1$, $\mu_1(x_1) = x_1$, and $\sigma_1(x_1) = \sigma_{\min}$. Notably, the conditional path is manually defined (e.g., function $u_t(x_1)$ and $\sigma_t(x_1)$).

## 2.2 INVERSE REINFORCEMENT LEARNING AND ADVERSARIAL IMITATION LEARNING

Inverse Reinforcement Learning derives a reward function given expert demonstrations, and optimize the policy based on this reward function (Ng et al., 2000). Belonging to IRL paradigm, adversarial imitation learning (AIL) reframes the problem through adversarial training. It employs a discriminator $D(s, a)$ to distinguish between expert and agent trajectories, while the policy $\pi$ is optimized to generate trajectories that fool the discriminator. The discriminator is optimized through the following objective:

$$\max_D \quad \mathbb{E}_{(s,a)\sim\rho_E}[\log D(s,a)] + \mathbb{E}_{(s,a)\sim\rho_\pi}[\log(1 - D(s,a))],$$

where $\rho_E$ and $\rho_\pi$ represent the state-action distributions of the expert and the learning agent respectively. The discriminator's output provides an adaptive reward signal that guides the policy optimization. Although the reward model in our method is grounded in the AIL paradigm, we refer to the overall approach as FM-IRL since we have an additional regularization module to stabilize the AIL training and policy learning.

# 3 METHODOLOGY

In this work, we propose FM-IRL, a teacher-student architecture where an FM model is trained to guide the online-update of agent's policy, given the expert demonstrations. Figure 1 illustrates the general workflow of FM-IRL, and Figure 7 in Appendix A shows the details of training procedure of both "teacher" FM model and "student" policy.

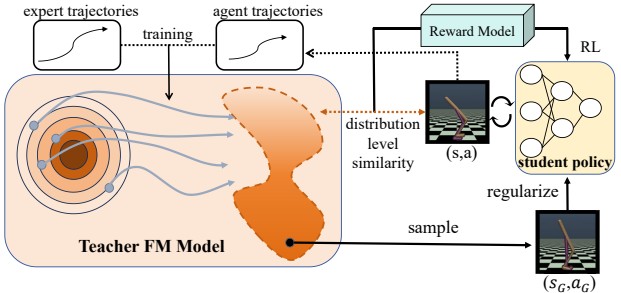

Figure 1: The FM model serves dual roles: (1) training a reward model for downstream reinforcement learning of the student policy, and (2) generating state-action pairs to regularize the student policy. The student policy interacts with the environment to collect data, which—along with expert data—is used to train the teacher FM model.

## 3.1 FLOW MATCHING AS AN EXPRESSIVE POLICY CLASS

We begin with how flow matching could be utilized to model the policy in RL. Flow Matching is known for its strong capability to fit complex data distributions, which is particularly essential in robot learning. This is because real-world demonstrations often exhibit multi-modal characteristics arising from diverse expert behaviors (Black et al., 2024). The FM policy is formulated via a conditional FM model defined by an ordinary differential equation (ODE):

$$\pi_\theta(a \mid s) = p_\theta(a_1 \mid s) = \int p_\theta(a_{0:T} \mid s)da_{0:T-1}, \tag{1}$$

where $p_\theta$ means the probability density function corresponding to specific vector field parametrized by $\theta$, $T$ means the number of discrete time steps used to approximate the continuous probability

flow, and $a_t$ denotes the status at flow time $t \in [0, 1]$, with $a_0 \sim \mathcal{N}(0, I)$ as the prior distribution and $a_1$ as the action used for the policy. The probability path is modeled through a vector field $v_\theta$ conditioning on $s$:

$$\frac{da_t}{dt} = v_\theta(a_t, s, t). \tag{2}$$

To train the FM policy, the most common approach is behavioral cloning (Torabi et al., 2018) based on the Flow Matching loss (Lipman et al., 2023):

$$\mathcal{L}_{\text{FM}}(\theta) = \mathbb{E}_{t \sim \mathcal{U}[0,1], (s,a_1) \sim \mathcal{D}_E, a_t \sim p_t(a|s,a_1)} \|v_\theta(a_t, s, t) - u_t(a_t \mid a_1, s)\|^2, \tag{3}$$

where $\mathcal{D}_E$ is the expert dataset, $u_t(a_t \mid a_1, s)$ is the pre-defined conditional flow velocity, and $a_t$ is sampled from the pre-defined conditional probability path connecting $a_0$ and $a_1$ given $s$. Upon sufficient training of the FM policy, actions are generated by solving the ODE backward in time:

$$a_1 = a_0 + \int_0^1 v_\theta(a_t, s, t)dt, \tag{4}$$

using numerical ODE solvers (e.g., Euler methods) (Euler, 1792).

Utilizing an FM policy as alternative of Gaussian policy in cloning expert behaviors has become promising due to its expressiveness in capturing the multi-modal expert behaviors. However, cloning the expert behaviors offline inherently lacks the online interaction, thus prohibits the agent's sufficient exploration with the environment and the ability of generalization. When the expert demonstration is limited or sub-optimal, this limitation is exacerbated to curb the generalization to unseen states. Unfortunately, updating the FM policy online via policy gradient-based methods remains difficult since the gradient of the FM policy's parameters is **hard to compute** (extremely unstable and costly). Appendix D provides theoretical insights of this limitation.

### 3.2 FM-ENHANCED DISCRIMINATOR

To address above-mentioned limitations, we aim to establish a framework (in imitation learning setting) where the agent benefits from the FM's distribution-matching (with expert data) capabilities and online policy optimization without suffering from the unstableness and cost of computing the FM policy gradient. Motivated by Adversarial Imitation Learning (AIL) (Ho & Ermon, 2016; Peng et al., 2018; Wang et al., 2024; Lai et al., 2024) that learns a discriminator to distinguish expert behavior and agent behavior and provides reward signals for the agent's online policy update, we propose to apply a "teacher" FM model to fit the expert state-action joint distribution and gain comprehensive understanding of the expert behavioral pattern. Then, we "infuse" the knowledge of "teacher" model into the reward function of an AIL discriminator. Specifically, we replace the traditional MLP-based discriminator in Eq. 5 by an FM-enhanced discriminator $D_{\text{FM}}$, and update the discriminator via:

$$\min_\theta \quad \mathcal{L}_{\text{FM}} = \mathbb{E}_{(s,a) \sim \rho_E}[\log(1 - D_{\text{FM},\theta}(s,a))] + \mathbb{E}_{(s,a) \sim \rho_\pi}[\log(D_{\text{FM},\theta}(s,a))]. \tag{5}$$

Ideally, the FM-enhanced discriminator could take state-action pair as input and output a scaler to answer the question: "*how possible that this action $a$ is sampled from expert data distribution given state $s$*". The traditional AIL's discriminator adopts an MLP architecture and naively models the data-point-level similarity (Wang et al., 2024). In contrast, the FM-enhanced discriminator models the distribution-level similarity between the agent's state-action pair and expert data distribution. In this way, the rich knowledge of FM model has been "infused" into the output of discriminator, which will be used to shape the reward signal by:

$$r_\theta(s, a) = \log(D_{\text{FM},\theta}(s,a)) - \log(1 - D_{\text{FM},\theta}(s,a)). \tag{6}$$

This framework fully follows the paradigm of AIL, and the reward function is adapted from AIRL (Fu et al., 2017), a popular variant of AIL, to provide a dense and guiding reward signal that encourages the agent's policy to match the expert's distribution during online update.

### 3.3 HOW TO DESIGN $D_{\text{FM},\theta}$?

The design philosophy of FM-enhanced discriminator $D_{\text{FM},\theta}$ is essential in our work since it directly affects the quality of reward signals in terms of the expressiveness of expert behavior. Inspired by

FPO (McAllister et al., 2025) which suggests that the loss value of an FM policy is a strong positive indicator of the similarity between one specific input and the target distribution, we adapt this insight into the design of $D_{\text{FM},\theta}$. Specifically, the "teacher" FM model takes the joint $(s, a)$ pair as input during training, modeling not only the action pattern but also the state distribution (the reason for such design will be explained in Appendix B). Then, the loss function is:

$$\mathcal{L}_{\text{FM}}(\theta) = \mathbb{E}_{t \sim \mathcal{U}[0,1], (s_1, a_1) \sim \mathcal{D}_E, (s_t, a_t) \sim p_t(\cdot | (s_1, a_1))} \left\| v_\theta((s_t, a_t), t) - u_t((s_t, a_t) \mid (s_1, a_1)) \right\|^2 . \quad (7)$$

Intuitively, the loss value will decrease as the FM model gradually fits the distribution of dataset $D$. Therefore, we can use the loss value to measure the distribution-level distance between one specific data point $(s', a')$ and the target distribution by canceling the uncertainty of the expectation that comes from $(s_1, a_1) \sim \mathcal{D}$ and replaces the $(s_1, a_1)$ by $(s', a')$ (McAllister et al., 2025):

$$Dist_\theta(s', a') = \mathbb{E}_{t \sim \mathcal{U}[0,1], (s_t, a_t) \sim p_t(\cdot | (s', a'))} \left\| v_\theta((s_t, a_t), t) - u_t((s_t, a_t) \mid (s', a')) \right\|^2 . \quad (8)$$

Meanwhile, inspired by (Lai et al., 2024) and to help the discriminator to discern the expert data and agent data, we further improve the design by conditioning the "teacher" FM model on an indicator variable $c$, which takes value from $\{0, 1\}$ to represent whether the FM model is fitting expert data or agent data:

$$Dist_\theta(s', a'|c) = \mathbb{E}_{t \sim \mathcal{U}[0,1], (s_t, a_t) \sim p_t(\cdot | (s', a'))} \left\| v_\theta((s_t, a_t), t|c) - u_t((s_t, a_t) \mid (s', a'), c) \right\|^2 , \quad (9)$$

where $Dist(s', a'|c = 1)$ represents the distance between $(s', a')$ and expert data distribution and $Dist(s', a'|c = 0)$ means the distance between $(s', a')$ and agent data distribution. Then, the FM-enhanced discriminator $D_{\text{FM},\theta}$ is formulated by Softmax transformation:

$$D_{\text{FM},\theta}(s, a) = \frac{exp(-Dist_\theta(s, a|c = 1))}{exp(-Dist_\theta(s, a|c = 1)) + exp(-Dist_\theta(s, a|c = 0))} . \quad (10)$$

This formulation helps to normalize the output value of discriminator within $[0, 1]$, making it compatible with the traditional AIL setting in Eq. 5. Also, it will give a positive indicator of distribution-level similarity between $(s, a)$ and expert data distribution, with the awareness of agent's data distribution.

## 3.4 POLICY REGULARIZATION

When only expert demonstrations are available and the true reward is unknown, online RL can suffer from reward mis-estimation on out-of-distribution state–action pairs (Fujimoto et al., 2019). This problem is amplified with an FM-enhanced discriminator: its strong expressiveness can easily overfit to expert data and generalize poorly, leading to misleading rewards on unseen states. Consequently, the agent may over-explore and struggle to accurately evaluate explored states, degrading learning performance. To improve the stableness of reward model training, we re-use the trained conditioned FM model to regularize the agent' policy. Specifically, we condition the FM model on $c = 1$ and let it generate state-action pairs that highly align with expert data distribution, and regularize the agent policy $\pi_\phi$ on the generated action to balance exploration and exploitation. The policy optimization objective is:

$$\max_\phi \quad \mathcal{J}(\phi) = \mathbb{E}_{(s,a) \sim \pi_\phi} \left[ \sum_k \gamma^k r_\theta(s, a) \right] - \beta \, \mathbb{E}_{(s_G, a_G) \sim G_\theta(\cdot | c=1), a_\pi \sim \pi_\phi} \left[ \left\| a_\pi - a_G \right\|^2 \right], \quad (11)$$

where the $G_\theta$ means the FM-model generator (which shares the same FM model as the FM-enhanced discriminator), $\gamma$ is the discount factor in RL context, $s_G, a_G$ means the generated state-action pair, and hyperparameter $\beta$ is for balancing the exploitation of expert data and exploration of online environment. Notably, $\pi_\phi$ represents the "student" policy parametrized by **simple MLP architecture**. Hence, the alternative perspective to interpret the second term is a more straight-forward way to "infuse" the "teacher" FM model's understanding of expert data distribution into the "student" MLP-based policy.

## 3.5 SUMMARY AND DISCUSSION

In general, FM-IRL contains **two** components that share **one** FM model: (1) an FM-enhanced discriminator with standard AIL workflow, and (2) a policy regularization module based on the FM model. Algorithm 1 in Appendix C shows the pseudo-code of the FM-IRL, Figure 7 in Appendix A shows the detailed architecture of our framework, Appendix E shows the implementation details, and we present the discussion of possible questions in a Q&A format in Appendix B.

# 4 EXPERIMENTS

As illustrated in Figure 2, we evaluate our method across six environments spanning navigation, locomotion, and manipulation. Each single experiment is repeated for 4 random seeds. Appendix E shows the hardware setup and more details.

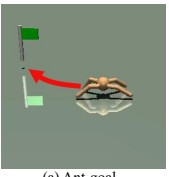 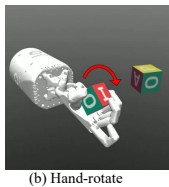 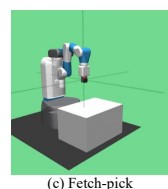 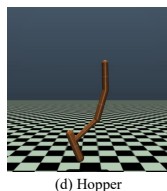 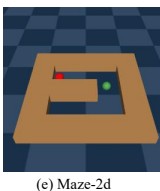 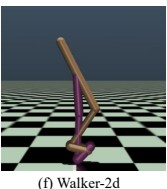

(a) Ant-goal    (b) Hand-rotate    (c) Fetch-pick    (d) Hopper    (e) Maze-2d    (f) Walker-2d

Figure 2: Overview of the six evaluation environments. *Navigation:*(a) **Ant-goal** tasks a quadruped agent with reaching a target position; (e) **Maze2d** requires an agent to navigate a 2D maze to a goal location; *Locomotion:* (d) **Hopper** requires fast and stable forward locomotion without falling; (f) **Walker2d** requires fast and stable forward locomotion without falling. *Manipulation:* (b) **Hand-rotate** requires dexterous in-hand rotation of a cube to a target orientation; (c) **Fetch-pick** requires grasping a block and placing it at a desired goal.

## 4.1 LEARNING EFFICIENCY STUDY

First, we evaluate the efficiency and effectiveness with which FM-IRL learns to perform the task from expert demonstrations. For tasks with a binary success indicator—including Ant-goal, Hand-rotate, Fetch-pick, and Maze2d—we report the training curve of success rate (y-axis, range 0–1) against training steps (x-axis). For tasks without a binary outcome measure, namely Hopper and Walker2d, we report the average return (y-axis, range 0 to $+\inf$) over training steps. To assess the learning efficiency of FM-IRL, we compare it with two categories of baselines: (1) Supervised Behavioral Cloning methods: Diffusion Policy (DP) (Chi et al., 2024) and Flow-Matching Policy (FP) (Zhang et al., 2025a); and (2) Inverse Reinforcement Learning (IRL) methods: GAIL (Ho & Ermon, 2016), VAIL (Peng et al., 2018), WAIL (Xiao et al., 2019), AIRL (Fu et al., 2017), and DRAIL (Lai et al., 2024). Note that methods in category (1) do not involve online policy updates, so their performance curves appear as horizontal lines. This comparison helps illustrate how FM-IRL addresses key limitations of FP (as well as DP). Comparisons with category (2) demonstrate the advantages of FM-IRL within the IRL paradigm. Please refer to Appendix G and G.1 for the technical details of baselines, the discussion about the difference between FM-IRL and DRAIL.

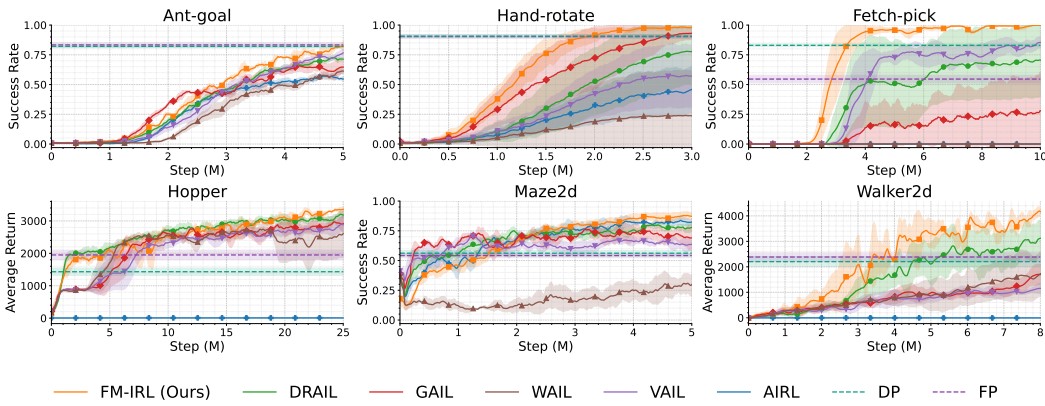

Figure 3: Learning curve of FM-IRL and baselines across 6 environments.

As shown in Figure 3, FM-IRL achieves the best final performance across all six environments except Ant-goal and reaches near-100% success in Hand-rotate and Fetch-pick. Compared to other IRL algorithms, FM-IRL converges faster and attains better ultimate performance in Hand-rotate,

Fetch-pick, and Walker2d, owing to its strong capacity to model complex expert distributions and maintain stable policy improvement during online updates. In the remaining three environments, it still outperforms other IRL methods by a smaller margin while exhibiting lower variance, indicating higher stability. Relative to supervised behavioral cloning methods (DP/FP), FM-IRL substantially surpasses them in all environments except Ant-goal, highlighting a fundamental limitation of Flow Matching Policy and Diffusion Policy—the lack of active exploration. FM-IRL addresses this through online reinforcement learning and a balanced exploration–exploitation strategy, enabling robust handling of unseen states. In Ant-goal, FM-IRL performs similarly to DP and FP because the task's simple path to the goal is fully covered by the expert data, making extensive exploration unnecessary. The quantitative results of this study is provided in Table 3 at Appendix F.1.

## 4.2 GENERALIZATION STUDY

To assess FM-IRL's ability to generalize to unseen states, we conducted a generalization study on **Manipulation** tasks (Figure 4 and Figure 5). Specifically, we introduced varying levels of noise to the initial and goal states in the Hand-rotate and Fetch-pick environment, testing whether the learned policies of FM-IRL and baseline methods could adapt to new trajectories under perturbed conditions. Noise scales included 1× (original expert setting), 1.25×, 1.5×, 1.75×, 2.00×, and 2.25×, where 1.25× denotes noise 1.25 times greater than that used during expert data collection. As shown in Figure 4, all methods exhibit performance degradation in Hand-rotate as noise increases, yet FM-IRL consistently outperforms the baselines across all noise levels. Specifically, FM-IRL maintains a success rate above 0.95 up to 1.5× noise, while the performance of DP and FP declines more sharply, dropping from 0.91 to nearly 0.82. This underscores the importance of online environmental interaction—even for policies capable of modeling complex action distributions. On the other hand, GAIL shows significant performance degradation (from around 0.91 to nearly 0.5) with only slight additional noise, and WAIL fails extensively under higher noise conditions. The limited generalization ability of GAIL and WAIL highlights the key role that FM played in capturing the multi-modal state-action distribution to handle diverse scenarios.

Figure 5 shows the final performance of each method after $10^7$ training steps in the Fetch-pick environment, with varying noise levels defined similarly as in Hand-rotate. We observe similar but more contrasting and compelling results: FM-IRL suffers from minimal and negligible performance degradation as noise levels increase. For baselines, static methods like FP and DP steadily suffer from performance drops due to the lack of generalization in behavior cloning. Notably, a large proportion of baselines fail to learn any patterns from demonstrations (i.e., achieve 0 success rate) as noise levels increase, such as GAIL, WAIL, and AIRL. The most competitive baselines in this environment, namely DRAIL and VAIL, can roughly maintain their performance when noise levels are not large (i.e., from 1.00 to 1.25), but suffer from drastic drops when noise levels further increase. Overall, FM-IRL demonstrates much stronger generalization to unseen states compared to all baselines in Fetch-pick, due to its policy expressiveness during training and robust online interaction with regularization. To further justify our claim, we also conducted the generalization study in **Navigation** and **Locomotion** tasks in Appendix F.3.

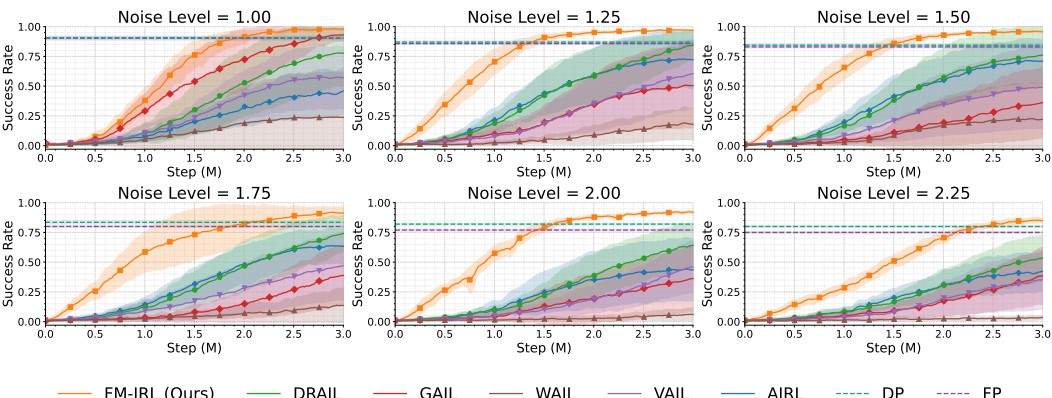

Figure 4: Learning curve of all methods in Hand-rotate environment across 6 noisy-levels.

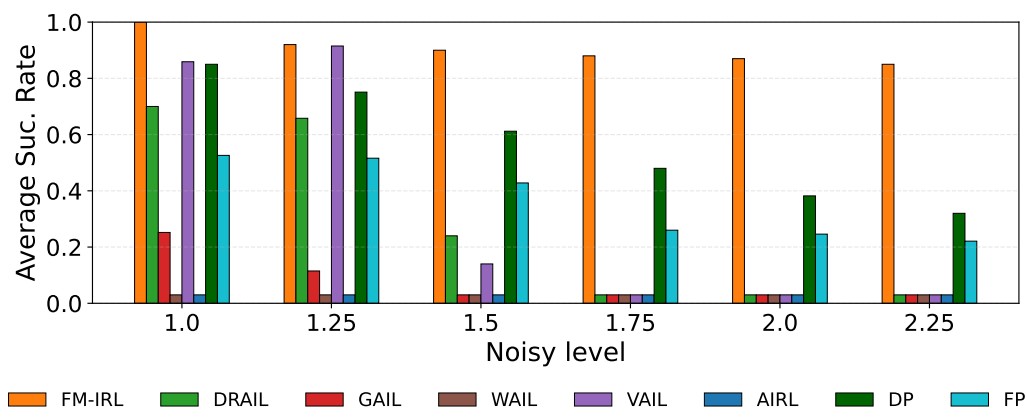

Figure 5: Performance of all methods in Fetch-pick environment across 6 noisy-levels.

## 4.3 ROBUSTNESS STUDY

As we previously explained, FM-IRL overcomes the limitation of lack-of-online-exploration inherent in traditional flow-matching and diffusion policies, which becomes catastrophic when expert demonstrations are sub-optimal. This section validates that claim by evaluating the robustness of FM-IRL, FP, and DP to sub-optimal expert data in the

Table 1: Robustness in Walker2d: We compare Diffusion Policy (DP), Flow-Matching Policy (FP), and our FM-IRL with sub-optimal experts, reporting average return (± std. over four seeds), with the best result highlighted in bold. The higher the ID values, the more optimal the expert data.

| ID | Expert Return | DP | FP | FM-IRL |
|---|---|---|---|---|
| 1 | 265.36 | **247.24** (±19.3) | 218.33 (±20.2) | 99.05 (±5.1) |
| 2 | 1965.58 | 1270.25 (±143.7) | 1342.47 (±168.1) | **2093.14** (±311.7) |
| 3 | 2662.14 | 1710.47 (±154.5) | 1578.91 (±175.6) | **2618.49** (±356.9) |
| 4 | 4653.69 | 1814.64 (±150.1) | 1746.66 (±173.3) | **4061.33** (±421.1) |
| 5 | 5357.37 | 2091.17 (±168.6) | 2014.28 (±199.7) | **4287.24** (±434.4) |

Walker2d environment. Specifically, we trained each algorithm using expert demonstrations with varying levels of episode return and compared the converged performance of the learned policies. As shown in Table 1, when expert returns are extremely low (ID: 1), DP slightly outperforms both FP and FM-IRL. This can be attributed to the fact that, with highly deficient expert data, FM-IRL struggles to infer a meaningful reward model, leading to largely meaningless online interactions. However, once the expert performance exceeds a novice level (i.e., achieves relatively higher returns), FM-IRL significantly surpasses both DP and FP (IDs: 2–5), and even achieves beyond-expert performance in some cases (IDs: 2–3). These results demonstrate the robustness of FM-IRL to sub-optimal expert data. In contrast, FP and DP, which lack online interaction, are prone to overfitting to the expert data. This is particularly detrimental when the data quality is poor. FM-IRL remains robust under the same conditions because the expert data only indirectly guides online exploration and policy updates through the reward signal: it influences the "teacher" FM model without directly constraining the agent's simple MLP policy.

## 4.4 CASE STUDY: FM POLICY WITH ONLINE REINFORCEMENT LEARNING

Given the poor robustness of FM policies in offline settings with sub-optimal expert data, a natural question is whether we can instead update an FM policy online with policy gradients. While theoretically feasible, this introduces substantial instability and extra computational overhead (see Appendix D). We empirically validate this in Maze2d by comparing FM-IRL's training curves and wall-clock time (Figure 6) against two naive baselines we proposed and one existing work that all couple online RL with an FM policy: FM-A2C (baseline), which updates the FM policy via an actor-critic framework using reparameterization without explicit density evaluation; FM-PPO (baseline), which applies Proximal Policy Optimization with reparameterization and requires explicit density computation for importance sampling; and FPO (McAllister et al., 2025), which uses the FM loss

to approximate a proxy importance ratio and then performs a standard PPO update.    As shown in

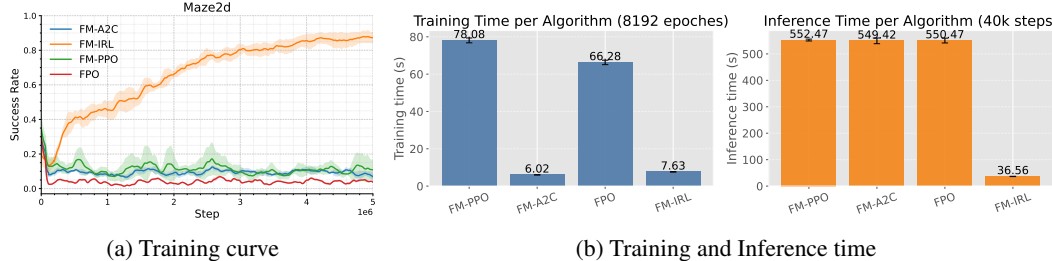

(a) Training curve  (b) Training and Inference time

Figure 6: Comparison of four algorithms for online updating FM policies

Figure 6a, three alternative ways for online updating FM policies via policy gradient fail to learn a feasible policy. Among these, FM-PPO appears to be the most promising one, reaching a peak success rate of near 0.33, but rapidly declines due to instability caused by backpropagation through time (BPTT). In contrast, FM-IRL avoids the instability of FM policy's gradient computation by infusing knowledge from a "teacher" FM model into a student policy. This policy can be updated stably using standard policy gradient methods, thereby achieving consistent and superior performance. Figure 6b compares the training time (for 8192 training epoches) and inference time (over 40k transition steps) of each algorithm. In terms of training cost, both FM-PPO and FPO are considerably more costly: FM-PPO requires Hutchinson trace estimation (Lipman et al., 2022) to approximate probability densities, while FPO must compute FM losses for both old and new policies to estimate importance sampling ratios for each sample—both of which are computationally intensive. In contrast, FM-IRL is significantly more efficient due to its architectural design that minimizes overhead. During inference, FM-PPO, FM-A2C and FPO have similarly high time cost since they use FM policies with the same architecture. In contrast, FM-IRL has significantly lower time cost. The key reason is that the other three methods adopt FM policy that requires multi-step numerical integration for action generation, while FM-IRL's behavioral policy is a simple MLP-based policy which is as capable as the FM policy in terms of performance.

It is worthy to note that, since FPO assumes the RL setting, namely we have access to the true reward of environment, we conduct the experiments of these three baselines (FPO, FM-A2C and FM-PPO) with the access to true reward. Although FM-IRL follows the IRL setting without access to the true reward, the comparison is still meaningful because all four methods perform online policy improvement driven by a reward signal. For fair comparison, we conducted additional experiments with the unified learned reward in Appendix F.4.

## 5 DISCUSSION

A insightful question of our paper will be: if our deployment policy uses a unimodal Gaussian head, how does multimodality improve policy expressiveness? Our key claim is that **multimodal training $\neq$ multimodal deployment**. A toy example will be helpful for understanding: consider expert demonstrations containing two distinct navigation strategies (modes) through a maze: left-first (40% of expert data), and right-first (60% of expert data). Traditional unimodal IRL methods (e.g., GAIL) employ simple discriminators that cannot capture fine-grained differences between these modes, resulting in coarse reward signals and potential **mode-averaging failures** (straight-first). In contrast, FM-IRL leverages multimodality through a fundamentally different mechanism: the FM teacher learns the complete multimodal structure of the expert distribution via flow matching, and this rich understanding is distilled into the discriminator through distribution-level distance metrics. When the student policy queries "Is action a good at state s?", the discriminator provides rewards informed by knowledge of all expert modes, recognizing which mode(s) the current behavior resembles the most and quantifying the similarity. Crucially, the student policy uses this multimodal-informed reward to discover **one optimal (single) mode** through online exploration, rather than naively averaging across incompatible modes. This is analogous to knowledge distillation (Hinton et al., 2015), where a complex ensemble teacher's soft outputs, which contains rich inter-class structure, enable a simpler student model to achieve comparable performance. Our FM-enhanced discriminator serves as such a "soft teacher," guiding the unimodal student toward one coherent mode while avoiding the pitfalls of mode collapse or averaging. We include more discussions in Appendix B.

## 6 RELATED WORK

Diffusion policy (Ankile et al., 2024; Chi et al., 2024; Pearce et al., 2023; Reuss et al., 2023; Sridhar et al., 2023; Ze et al., 2024) pioneers the use of modern generative models for policy representation, advancing the ability to capture complex action distributions beyond Gaussian approximations. Following the success of diffusion models, Flow Matching (FM) offers a promising alternative that enables efficient training, fast sampling, and improved generalization compared to diffusion models (Lipman et al., 2023; Zheng et al., 2023), and has gained popularity in robot learning (Braun et al., 2024; Zhang & Gienger, 2025), image generation (Lipman et al., 2023; Esser et al., 2024), and video synthesis (Kong et al., 2025; Wan et al., 2025a).

However, traditional diffusion or FM policies are typically designed for offline settings to clone expert data through supervised learning. Recently, several works have attempted to adapt diffusion-based policies to reinforcement learning beyond mere behavioral cloning. For example, DQL (Wang et al., 2023) and IDQL (Hansen-Estruch et al., 2023) employ diffusion models to represent the policy network and use an actor-critic architecture (Konda & Tsitsiklis, 1999) to perform policy-gradient optimization. Nevertheless, these methods suffer from extreme instability during policy gradient computation due to the long-chain structure of the diffusion policy. FQL (Park et al., 2025) follows a similar approach but uses an FM policy and infuses knowledge from a full FM model into a simpler one to improve stability. However, these methods are designed only for offline settings and lack support for exploration in online environments.

To adapt the strong distribution-matching capability of diffusion or FM policies to online settings, DIPO (Yang et al., 2023) introduces action gradients to directly update action batches based on the Q-function and re-fits the distribution of new action batches, but is limited to purely value-based reinforcement learning. Alternatively, QSM (Psenka et al., 2025) and DPPO (Ren et al., 2024) use policy gradients to fine-tune pre-trained diffusion policies in online environments. Similar to DPPO, ReinFlow (Zhang et al., 2025b) pre-trains a flow-matching policy and fine-tunes it in online environment. Flow-GRPO (Liu et al., 2025) and ORW-CFM-W2 (Fan et al., 2025) also use RL to fine-tune flow matching models, but with a focus on vision-based tasks rather than policy modeling. Our method eliminates the need for post-training fine-tuning and enables learning a policy from scratch that benefits from the distribution-matching capability and efficiency of Flow Matching Model.

Similar to ours, many IRL methods learn a reward to guide online exploration. The most established is the GAIL family (Ho & Ermon, 2016)—including GAIL-GP, VAIL, and WAIL (Peng et al., 2018; Xiao et al., 2019)—which pioneered adversarial imitation learning. However, GAIL-style methods typically use simple discriminators that do not explicitly model distributions, often yielding imprecise rewards for behaviors close to expert demonstrations. To address this, DiffAIL (Wang et al., 2024) and DRAIL (Lai et al., 2024) adopt diffusion-based discriminators to improve distribution matching capabilities. Despite better discrimination, they are computationally inefficient and tend to poorly calibrate rewards on out-of-distribution (OOD) state-action pairs. Our FM-IRL overcomes these issues by using an optimal-transport Flow Matching discriminator that is significantly more efficient. We further add an regularization (infusing) term that transfers the discriminator's rich expert-distribution knowledge directly to the policy, yielding more stable and accurate reward guidance. This regularization also penalizes excessive exploration of OOD states, mitigating catastrophic misestimation and better balancing exploration and exploitation. Table 6 in Appendix H summarizes the properties of existing works and highlights the design philosophy of our FM-IRL.

## 7 CONCLUSION

In this work, we identify the key limitations of behavior-cloning-based FM policies, and the key challenges of applying FM policies in online reinforcement learning. Then, we propose a novel framework, Flow Matching Inverse Reinforcement Learning (FM-IRL), that enables effective online policy updates using expert demonstrations while enabling the policy to benefit from the strength of FM. Extensive experiments across robot navigation, manipulation, and locomotion tasks validate the effectiveness, generalizability, robustness, and efficiency of the proposed approach. The framework holds potential for extension to a broader range of demonstration-based learning applications, such as LLM-based agents and autonomous driving.

ETHICS STATEMENT

This study complies with the ICLR Code of Ethics and relevant research integrity guidelines. The dataset used in this work is publicly available and does not contain private or sensitive information. We have conducted a bias and fairness analysis of our methodology, and results indicate no significant discriminatory impact. Potential risks associated with the proposed applications have been evaluated. The funding source of this paper had no influence on the study's design or outcomes. All authors confirm the originality of the work.

REPRODUCIBILITY

To ensure the reproducibility of the results presented in this paper, we have included a detailed description of the algorithm in Appendix C. Furthermore, comprehensive implementation details along with a description of our hardware environment are provided in Appendix E. The authors undertake to make the complete codebase open-source upon the formal acceptance of this work.

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

## A    DETAILED FRAMEWORK ILLUSTRATION

Figure 7 illustrated the detailed architecture of FM-IRL, and shows the training procedure of both "teacher" FM Model and "Student" MLP policy.

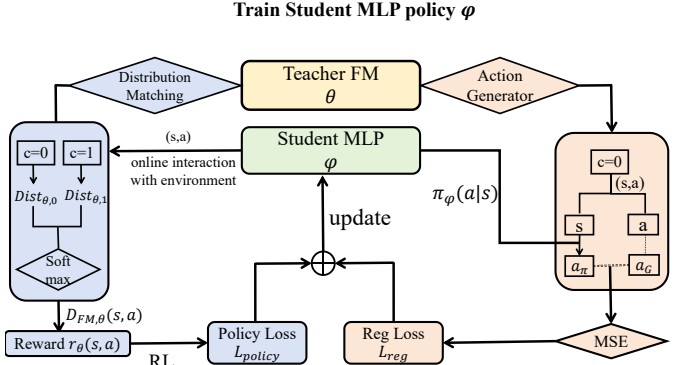 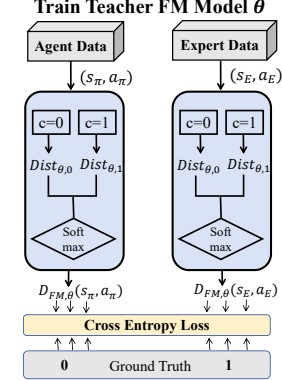

Figure 7: Detailed architecture of FM-IRL. Left: the "teacher" FM model "infuse" the knowledge of expert data distribution to the student MLP-based policy via two channels: the reward function in RL algorithm and the policy regularization module. Right: the training procedure of teacher FM model is adapted from standard AIL paradigm.

## B    DISCUSSION

This sections provides the answers to possible questions about FM-IRL framework and the experiment results in Q&A format below.

### B.1    QUERY ABOUT EXPERIMENT RESULTS

**QA**

**Q. Why the performance improvement of FM-IRL is marginal in Navigation tasks?**

**A.** This observation exactly aligns with the motivation of FM-IRL. Please note that the advantage of our FM-enhanced discriminator lies in capturing multi-modal expert distributions during training, and our regularization term aims to stabilize the training process. In navigation tasks, the objective is to move the agent towards a target position. We explained in the experiment session that the success trajectory is often unique (uni-modal) and is fully covered by expert data, obviating the need of excessive exploration and complex, multi-modal training. Therefore, FM-IRL shows marginal improvement but the experiments also proves in such relatively "uni-modal tasks", FM-IRL roughly degraded to other baselines but with greater stability (reflected by lower training variance), preserving its lower-bound performance. In Manipulation tasks like Hand-rotate and Fetch-pick, the goal is not anymore "moving to a target place" but to try to manipulate an object, enabling more complexity, flexibility and multi-modality in expert data. In such tasks, FM-IRL significantly outperforms other methods, validating our motivation.

**Q. The experiment result of robustness study (section 4.3) does not support the claim that FM-IRL overcomes the limitation of suboptimal expert data, as they are approximately or often below the expected return of demonstration data.**

**A.** Actually, the claim of "robustness to sub-optimal data" is relative to baseline, rather than expert performance. Since our method is inherently imitation learning, the policy is optimized absolutely from expert data. Therefore, the return of learned policy can hardly surpass expert demonstration. However, when experts are sub-optimal, our method exhibits

much stronger robustness compared to DP and FP, highlighting the importance of online exploration which forms the motivation of our method.

## B.2 QUERY ABOUT FM-IRL FRAMEWORK

**QA**

**Q. Why do you use a complex FM model as the "teacher" but only use a simple MLP-based policy as the "student"?**

**A.** This design leverages the advantages of the FM model in policy representation while addressing the limitations of directly applying FM to policy modeling. Specifically, the FM policy is hard to update using online RL algorithms due to its architecture, whereas our "student" MLP policy can explore the environment and be updated online efficiently thanks to its simple structure. Furthermore, by infusing the knowledge of the "teacher" FM model into the "student" MLP policy, the agent benefits from both fast inference and an awareness of the expert data distribution.

**Q. Why do you model the joint distribution of $s$ and $a$ in the "teacher" FM model, rather than modeling the conditional distribution $a \mid s$ as in the traditional FM policy paradigm?**

**A.** First, the "teacher" FM model is used only for training the "student" policy, where we only need state–action pairs and thus do not need to generate actions conditioned on state at test time. More importantly, based on Eq. 11, modeling the joint state–action distribution facilitates state regularization to penalize over-exploration of unseen states where the value function is hard to estimate. This is because we need to sample from the FM generator when computing the regularization term; if we model the action conditioning on the state, we can sample actions only for visited states, which regularizes expert actions but ignores expert states.

**Q. How to understand "infuse" in FM-IRL?**

**A.** The two components of FM-IRL both play a role in "infusing" in different ways. (1) The FM-enhanced discriminator **indirectly** infuses the FM model's knowledge to the RL agent through the **medium** of the reward function. (2) The regularization term **directly** performs "infusing" by minimizing the MSE loss between the agent's actions and the actions generated by the FM model. However, they collectively "teach" the "student" policy about the distribution of expert data.

**Q. Since one of your key contribution is an FM-enhanced discriminator, why you name your framework "FM-IRL" instead of "FM-AIL"?**
**A.** Indeed, our method employs adversarial training, which might suggest the name "FM-AIL." However, we deliberately chose "FM-IRL" to reflect our adherence to the classical two-stage IRL paradigm: (1) learning a reward function $r_\theta(s, a)$ from expert demonstrations $\mathcal{D}_E$, and (2) optimizing a policy $\pi_\phi$ using the learned reward via RL. While AIL, specifically GAIL (Ho & Ermon, 2016), demonstrates that AIL is equivalent to IRL with entropy regularization, our method extends beyond standard AIL in a key aspect. Standard AIL derives rewards solely from discriminator outputs for occupancy measure matching. In contrast, FM-IRL incorporates a **dual mechanism**: (1) an FM-enhanced discriminator for reward modeling, and (2) explicit FM-based policy regularization. This second component goes beyond the conventional AIL framework and represents a distinct contribution that leverages FM's generative modeling capacity.

**Q. Does online interaction truly provide benefits when rewards come solely from offline data? Since the reward function is learned from an offline dataset, it can only encourage in-distribution behavior, and regularization further restricts exploration—making the framework essentially mimic the offline dataset through joint optimization.**

**A.** While the reward model trains on offline data, it learns a **generalizable function** that evaluates arbitrary $(s, a)$ pairs, including unseen ones. Consider RLHF: it learns from a fixed human feedback dataset yet enables models to generalize to novel prompts (Ouyang et al., 2022). The reward function is a learned abstraction capturing what makes behavior "expert-like," not merely memorizing specific $(s, a)$ pairs. Saying we "mimic the offline dataset" oversimplifies what's happening. We're learning the *expert's task objective*—fundamentally different from imitation. Behavioral cloning methods (Diffusion Policy, Flow Matching Policy) directly memorize $(s, a)$ mappings with poor generalization. GAIL matches state-action occupancy measures point-wise, still limiting generalization. FM-IRL

learns the *distributional structure* of expert data, enabling the policy to understand what makes expert behavior successful and whether new $(s, a)$ pairs align with this objective. Empirically, our learned policies sometimes *exceed expert performance*, showing we've captured something deeper than surface behavior, likely because expert demonstrations contain suboptimal actions due to human error or limited data. FM captures the "general strategy" underlying expertise. On regularization: it doesn't exacerbate distribution-matching. It **constrains exploration** to regions where reward estimates remain reliable, preventing the agent from venturing into highly out-of-distribution states where the learned reward becomes uninformative.

## C  PSEUDO CODE

This section shows the implementation Pseudo Code of FM-IRL algorithm.

---

**Algorithm 1 Flow-Matching Inverse Reinforcement Learning (FM-IRL)**

---

1: **Input:** Expert trajectory dataset $D_E$, initial agent policy $\pi_\theta$, initial FM generator $G_\theta$
2: **Initialize:** Student policy parameters $\phi$, FM generator parameters $\theta$
3: **while** not converged **do**
4:     Sample expert trajectories $\tau_E \sim D_E$
5:     Roll out agent policy $\pi_\theta$ to obtain trajectories $\tau_A$

6:     **Student Policy Update:**
7:         Compute reward $\nabla_\theta(s, a)$ for $(s, a)$ in $\tau_A$ via Eq. 6
8:         Compute policy loss $\mathcal{J}(\phi)$ based on policy optimization objective (Eq. 11)
9:         Update policy parameters:
10:            $\phi \leftarrow \phi + \alpha \nabla_\phi \mathcal{J}(\phi)$

11:    **Teacher FM Model Update:**
12:        Compute FM discriminator loss $\mathcal{L}_{\text{FM}}(\theta)$ based on Eq. 5, 9 and 10
13:        Update FM generator parameters:
14:            $\theta \leftarrow \theta - \gamma \nabla_\theta \mathcal{L}_{\text{FM}}(\theta)$
15: **end while**
16: **Output:** Optimized agent policy $\pi_{\phi^*}$ and FM generator $G_{\theta^*}$

---

## D  WHY AN FM POLICY IS HARD TO UPDATE IN THE TRADITIONAL POLICY–GRADIENT PARADIGM

In this section, we will provide mathematical and statistical insight for readers to understand why FM policy is hard to update in online reinforcement learning via the traditional policy-gradient paradigm.

**Setup.**  Let $\pi_\theta(a \mid s)$ be a stochastic policy. We define the objective as

$$J(\theta) \;=\; \mathbb{E}_{(s,a)\sim\pi_\theta}\big[\, Q(s, a)\,\big], \tag{12}$$

where $Q(s, a)$ denotes a state–action value (critic). In the policy–gradient framework we are interested in its gradient

$$\nabla_\theta J(\theta) \;=\; \nabla_\theta \, \mathbb{E}_{(s,a)\sim\pi_\theta}\big[\, Q(s, a)\,\big]. \tag{13}$$

Since $\theta$ appears inside the sampling distribution $\pi_\theta$, this gradient is not straightforward to compute. In the traditional case, three standard approaches are commonly used: (1) likelihood–ratio, (2) explicit reparameterization (pathwise), and (3) implicit reparameterization (via the CDF). We briefly recall them, then explain why each becomes difficult for a Flow–Matching (FM) policy.

**Route I: Likelihood–ratio (LR).**

$$\nabla_\theta J(\theta) \;=\; \mathbb{E}[\nabla_\theta \log \pi_\theta(a \mid s)\, Q(s, a)]. \tag{14}$$

For a Gaussian with mean $\mu_\theta(s)$ and covariance $\Sigma_\theta(s)$, the log-likelihood can be written in closed form as

$$\log \pi_\theta(a \mid s) = -\frac{1}{2}\left((a - \mu_\theta)^\top \Sigma_\theta^{-1}(a - \mu_\theta) + \text{logdet}(2\pi\Sigma_\theta)\right), \tag{15}$$

so both $\log \pi_\theta$ and $\nabla_\theta \log \pi_\theta$ are easy to compute and numerically stable.

**Route II: Explicit reparameterization (pathwise).** Introduce a standard Gaussian noise $\xi \sim \mathcal{N}(0, I)$ with the same dimension as the action. The action can then be written as

$$a = g_\theta(s, \xi) = \mu_\theta(s) + L_\theta(s)\,\xi, \qquad L_\theta(s)\,L_\theta(s)^\top = \Sigma_\theta(s), \tag{16}$$

With this reparameterization, we have

$$\nabla_\theta J(\theta) = \mathbb{E}_{s,\xi}\left[\nabla_a Q(s, a)\,\frac{\partial g_\theta(s, \xi)}{\partial \theta}\right]. \tag{17}$$

Since $g_\theta$ is affine in the Gaussian case (a linear transform plus a shift), the backpropagation path involves only a single matrix–vector multiplication and addition. As a result, the gradient is inexpensive to compute and numerically stable.

**Route III: Implicit reparameterization gradients (IRG).** When a distribution does not have a simple reparameterization map $g_\theta(s, \xi)$, one can instead differentiate through its cumulative distribution function (CDF) $F_\theta$. The idea is to use the inverse-CDF sampling scheme: draw $u \sim \text{Unif}[0, 1]$ and define the sample $a$ implicitly by

$$F_\theta(a) = u.$$

Since $u$ is fixed once drawn, differentiating this relation with respect to $\theta$ (via the implicit function theorem) yields, in one dimension,

$$\frac{\partial a}{\partial \theta} = -\left(\frac{\partial F_\theta}{\partial a}\right)^{-1}\frac{\partial F_\theta}{\partial \theta}. \tag{18}$$

Here $\frac{\partial F_\theta}{\partial a}$ is exactly the density $p_\theta(a)$ evaluated at the sample, while $\frac{\partial F_\theta}{\partial \theta}$ captures how the CDF changes as the parameters $\theta$ vary. Intuitively, as $\theta$ changes, the sampled point $a$ must shift in order to preserve the same CDF value $u$.

In higher dimensions, a single CDF is not enough; instead one introduces a sequence of conditional CDFs, known as the *Rosenblatt transform*, to map a vector of uniform random variables into a valid sample. This provides a general way to obtain reparameterization gradients even for distributions that lack an explicit sampler. In practice, however, for distributions like the Gaussian, this route is rarely used because an explicit affine reparameterization is already available (Route II).

**FM policies in a line.** An FM policy samples actions by integrating a conditional ODE flow:

$$\frac{dx_t}{dt} = v_\theta(x_t, s, t), \quad x_0 = \xi \sim p_0, \quad a = x_1. \tag{19}$$

Let $a = \Phi_{0\to1}^{\theta,s}(\xi)$ so that $\pi_\theta(\cdot \mid s) = (\Phi_{0\to1}^{\theta,s})_\# p_0$ (the pushforward of $p_0$ by the flow map). Sometimes the flow is trained with an explicit density (a CNF), sometimes by velocity regression without a likelihood ("pure FM").

WHY THE THREE TRADITIONAL ROUTES BECOME DIFFICULT FOR FM POLICY

**LR for FM: likelihoods are no longer easy to compute.** For CNFs, the log–density along a trajectory satisfies

$$\frac{d}{dt}\log p_t(x_t) = -\text{tr}\left(\frac{\partial v_\theta}{\partial x}(x_t, s, t)\right), \tag{20}$$

$$\implies \log \pi_\theta(a \mid s) = \log p_0(\xi) - \int_0^1 \text{tr}\left(\frac{\partial v_\theta}{\partial x}(x_t, s, t)\right) dt, \qquad a = \Phi_{0\to1}^{\theta,s}(\xi). \tag{21}$$

Thus, to obtain $\log \pi_\theta(a \mid s)$ for one sample we must (i) solve the ODE to obtain the path $x_{0:1}$, (ii) compute a trace integral along that path (often with a Hutchinson estimator, which needs multiple Jacobian–vector or vector–Jacobian products), and then (iii) differentiate this pipeline with respect to $\theta$. Since LR needs $\nabla_\theta \log \pi_\theta$, it inherits the same pipeline. Let $N_{\text{fe}}$ be the average number of vector–field evaluations per ODE solve, $K$ the number of probe vectors for the trace, $B$ the batch size, and $H$ the rollout horizon. A single on–policy update already needs roughly

$$\Omega\big(B\,H\,N_{\text{fe}}\,(1+K)\big) \tag{22}$$

forward evaluations, before parameter backpropagation. In "pure FM", $\log \pi_\theta$ is not available at all, so LR is inapplicable.

**IRG for FM: the needed CDFs are not exposed.** Implicit reparameterization requires access to $F_\theta$ (and, in multiple dimensions, conditional CDFs). An FM policy provides an implicit sampler $a = \Phi_{0 \to 1}^{\theta,s}(\xi)$, not a tractable $F_\theta$. Forcing a CNF only to recover $F_\theta$ brings us back to the same ODE solves and trace integrals as LR. The obstacle is structural.

**Pathwise for FM: reasonable in form, but it brings BPTT.** Formally, FM policies are already reparameterized. Given noise $\xi \sim p_0$ and flow map $\Phi_{0 \to 1}^{\theta,s}$, the action is $a = \Phi_{0 \to 1}^{\theta,s}(\xi)$. The objective and its gradient can be written as

$$J(\theta) = \mathbb{E}_{s,\xi}\left[Q\Big(s, \Phi_{0 \to 1}^{\theta,s}(\xi)\Big)\right], \tag{23}$$

$$\nabla_\theta J(\theta) = \mathbb{E}_{s,\xi}\left[\nabla_a Q(s,a)\, \frac{\partial \Phi_{0 \to 1}^{\theta,s}(\xi)}{\partial \theta}\right], \qquad a = \Phi_{0 \to 1}^{\theta,s}(\xi). \tag{24}$$

Thus the central difficulty is computing the sensitivity $\partial\Phi/\partial\theta$, i.e. how the final action changes when the policy parameters change. There are two standard approaches:

*(i) Backpropagation Through Time (BPTT).* Here one treats the numerical ODE solver as a sequence of discrete updates

$$x_{k+1} = \Phi_k(x_k; \theta), \qquad k = 0, \ldots, N-1, \qquad a = x_N,$$

and attaches a terminal loss $L = \ell(a)$. Gradients are then propagated backward step by step, like training a recurrent neural network:

$$\lambda_N = \nabla_{x_N}\ell, \qquad \lambda_k = \left(\tfrac{\partial \Phi_k}{\partial x_k}\right)^\top \lambda_{k+1}.$$

The overall parameter gradient accumulates as

$$\nabla_\theta L = \sum_{k=0}^{N-1} \left(\tfrac{\partial \Phi_k}{\partial \theta}\right)^\top \lambda_{k+1}.$$

*(ii) Continuous adjoints / sensitivities.* Instead of unrolling the solver, one can differentiate the continuous ODE system directly. In the forward sensitivity method, one integrates the matrix

$$S_t = \frac{\partial x_t}{\partial \theta}, \quad \frac{dS_t}{dt} = \frac{\partial v_\theta}{\partial x}(x_t, s, t)\, S_t + \frac{\partial v_\theta}{\partial \theta}(x_t, s, t), \quad S_0 = 0,$$

and obtains $\partial a/\partial\theta = S_1$ at the end of the flow. Alternatively, the adjoint method integrates a backward variable $\lambda_t$, yielding

$$\nabla_\theta J(\theta) = \mathbb{E}\left[\int_0^1 \left(\tfrac{\partial v_\theta}{\partial \theta}(x_t, s, t)\right)^\top \lambda_t\, dt\right].$$

Both formulations are mathematically equivalent; the choice depends on whether one prefers forward or backward accumulation of sensitivities.

**Why this still struggles in practice.** Compared to the Gaussian case, computing sensitivities for FM policies is significantly more involved because the gradient path runs through an entire ODE solver rather than a short affine transformation.

*Cost.* Each sample requires both a forward ODE solve and a backward pass of similar or higher cost. With adaptive solvers or mildly stiff dynamics, the average number of function evaluations $N_{\text{fe}}$ grows, making training expensive. The complexity of one update scales roughly as

$$\widetilde{O}\big(B\, H\, N_{\text{fe}}\, C_{\text{jvp/vjp}}\big),$$

where $B$ is the batch size, $H$ is the rollout horizon, and $C_{\text{jvp/vjp}}$ is the cost of a Jacobian–vector or vector–Jacobian product. Continuous adjoints save memory but not time; in stiff regimes, the backward solve can even be harder than the forward one.

*Numerical mismatch.* Continuous adjoints differentiate the *continuous* ODE, while the forward pass uses a *discretized* solver. With adaptive step sizes or differing time grids, the two gradients can diverge, hurting stability. Using discrete adjoints (matching the backward steps to the forward solver) or checkpointing can reduce this mismatch, but both add implementation effort and runtime overhead.

*Variance.* The gradient estimator combines the critic's action gradient $\nabla_a Q(s, a)$ with the flow sensitivity $\partial a / \partial \theta$. For actions generated by a deep ODE chain, the sensitivity can be large and noisy, amplifying variance and making optimization unstable. In practice this often necessitates smaller learning rates, stronger target networks, and additional regularization, which lowers sample efficiency.

**Conclusion.** In Gaussian policies, likelihoods are easy to compute, the reparameterization path is short, and CDFs are tractable, so all three routes are practical. In FM policies, however, LR requires ODE solves and trace integrals, IRG depends on CDFs that are not exposed, and the pathwise route—though natural in form—forces gradients through an ODE solver, leading to high cost, numerical issues, and higher variance. The gradients do exist, but in high-dimensional, long-horizon, online settings they are rarely a favorable trade-off in practice.

# E  IMPLEMENTATION DETAILS

The experiment environments are customized and adapted from widely used platforms, including OpenAI Gymnasium, (Towers et al., 2024), D4RL (Fu et al., 2021), and MuJoCo (Todorov et al., 2012). Part of the baseline RL, IL, and IRL implementations are adapted from rl-toolkit (Szot, 2024). The implementation of noisy environment in section 4 are adapted from Goal-prox-il (Lee et al., 2021) and DRAIL (Lai et al., 2024).

All experiments are conducted on a Linux server equipped with four NVIDIA A40 (48GB) GPUs and an AMD EPYC 7543P 32-core CPU.

We show the algorithmic and experimental implementation details below.

## E.1  ALGORITHMIC DETAILS

### E.1.1  CHOICE OF CONDITIONAL PROBABILITY PATHS IN FM

We model the joint vector $x = (s, a)$ and condition the model on $c \in \{0, 1\}$ (e.g., $c{=}1$ for expert data, $c{=}0$ for agent data). For the conditional probability path, we use a simple straight-line (i.e. Optimal Transport path (Lipman et al., 2022)) conditional path between a noise start point and a target joint sample:

$$x_0 \sim \mathcal{N}(0, I), \qquad x_t = (1 - t)\, x_0 + t\, x_1, \quad t \in [0, 1], \quad x_1 = (s_1, a_1). \tag{25}$$

Under this path, the target velocity is time-constant:

$$u_t(x_t \mid x_1, c) = x_1 - x_0, \tag{26}$$

while the model predicts $v_\theta(x_t, t \mid c)$. The corresponding Flow Matching loss is:

$$\mathcal{L}_{\text{FM}}(\theta) = \mathbb{E}_{\substack{(x_1, c) \sim \mathcal{D},\, x_0 \sim \mathcal{N}(0, I) \\ t \sim \mathcal{U}[0, 1]}} \left\| v_\theta(x_t, t \mid c) - (x_1 - x_0) \right\|^2, \qquad x_t = (1 - t)x_0 + tx_1. \tag{27}$$

Optimal Transport paths offer key advantages in probability path construction. First, OT paths form the shortest geodesic connections between distributions, mathematically achieved by minimizing the transport cost. This shortest-path property yields two central benefits: it significantly improves training efficiency, as straighter trajectories result in lower variance and faster convergence; and it enhances sample quality by better preserving structural features of the target distribution.

### E.1.2 COMPUTATION OF $Dist$ IN REWARD MODEL

We use the FM model over the joint input $x = (s, a)$ and condition on $c \in \{0, 1\}$ (expert: $c=1$, agent: $c=0$). Given a target pair $x_1 = (s, a)$, we sample a noise start $x_0 \sim \mathcal{N}(0, I)$ and define the straight-line conditional path

$$x_t = (1 - t)x_0 + tx_1, \qquad t \in [0, 1], \tag{28}$$

with target velocity

$$u_t(x_t \mid x_1, c) = x_1 - x_0. \tag{29}$$

The FM distance (our "loss") at $(s, a)$ under condition $c$ is the expectation of the per-sample discrepancy:

$$Dist_\theta(s, a \mid c) = \mathbb{E}_{t \sim \mathcal{U}[0,1], \, x_0 \sim \mathcal{N}(0,I)}\Big[\big\| v_\theta(x_t, t \mid c) - (x_1 - x_0)\big\|_2^2\Big], \quad x_t = (1-t)x_0 + tx_1. \tag{30}$$

**Monte Carlo estimation.** We approximate the expectation by Monte Carlo. With $S$ samples $\{(t_i, x_0^{(i)})\}_{i=1}^S$,

$$\widehat{Dist}_\theta(s, a \mid c) = \frac{1}{S} \sum_{i=1}^S \big\| v_\theta\big(x_t^{(i)}, t_i \mid c\big) - \big(x_1 - x_0^{(i)}\big)\big\|_2^2, \qquad x_t^{(i)} = (1 - t_i)x_0^{(i)} + t_i x_1. \tag{31}$$

In practice we use stratified time sampling $t_i \in [0, 1]$ with small jitter and vectorize all $S$ samples across the batch for efficiency. For the discriminator branch during training, we also support the $S=1$ single-sample variant (one $t$ and one $x_0$ per $(s, a)$) for speed, which is an unbiased estimator of the expectation.

**From $Dist$ to reward.** Given the label-conditioned distances, we form the FM-enhanced discriminator via a Softmax over negative distances:

$$D_{\text{FM},\theta}(s, a) = \frac{\exp\big(-\widehat{Dist}_\theta(s, a \mid c{=}1)\big)}{\exp\big(-\widehat{Dist}_\theta(s, a \mid c{=}1)\big) + \exp\big(-\widehat{Dist}_\theta(s, a \mid c{=}0)\big)}, \tag{32}$$

and compute the reward with the standard AIL/AIRL transform

$$r_\theta(s, a) = \log D_{\text{FM},\theta}(s, a) - \log\big(1 - D_{\text{FM},\theta}(s, a)\big). \tag{33}$$

Thus, smaller FM loss implies smaller $Dist$, larger $D_{\text{FM}}$, and a higher reward.

### E.1.3 IMPLEMENTATION PHILOSOPHY OF FM-IRL

A notable strength of FM-IRL lies in its algorithmic elegance: it's reward model is designed entirely within the GAIL framework while addressing the limitations of traditional GAIL-family methods. This architectural choice ensures that FM-IRL can be implemented with minimal modifications to standard GAIL codebases. Specifically, practitioners only need to implement the computation details of $D_{\text{FM},\theta}$ while maintaining interface compatibility with GAIL's discriminator API. The rest of the training pipeline—policy optimization, replay buffer management, and adversarial updates—remains identical to vanilla GAIL.

In addition to the reward model, when the regularization weight $\beta = 0$, the training of the student MLP policy strictly follows the standard paradigm of PPO, including learning a value function/advantage function (critic network) based on reward model, modeling the policy as MLP with Gaussian head, and performing trust-region policy optimization based on PPO-objective. When $\beta \neq 0$, we only need to implement an additional regularization term. This means the implementation of the student MLP policy could also be built on PPO implementation, with slight modification of the reward function and an addition regularization module.

These strength collectively makes FM-IRL highly accessible and easy to implement.

### E.1.4 Policy Architecture Details

For all methods that use MLP policy architecture, we adopts a diagonal Gaussian policy: the policy is a diagonal Gaussian with a state-dependent learnable mean $\mu$ and single learnable log-std vector $\Sigma$ of size *action_dim*. $\Sigma$ is a global **state-independent** parameter, initialized to zeros (so std=exp(0)=1), broadcast across the batch, and optimized jointly with the actor via the PPO objective; it receives gradients from both the policy log-likelihood term and the entropy bonus in standard PPO paradigm. It is not a fixed constant and not state-dependent. Using a single global learnable log-std vector for PPO's diagonal Gaussian policy improves stability and simplicity: it decouples exploration scale from state, reducing gradient variance, typically yielding more stable, reproducible training and smoother convergence under PPO's clipped objective compared to state-dependent variance heads.

### E.2 Experimental Details

To ensure fair comparison, we adopt a practical two-tier strategy for hyperparameter configuration. For shared components, like policy and critic networks, learning rates, batch sizes, and discount factors, all methods use **identical settings**. This ensures that performance differences reflect genuine algorithmic improvements rather than implementation advantages. For FM-IRL's unique components (FM-enhanced discriminator), we deliberately choose **one straightforward set of hyperparameters and fix them across all tasks** without per-task tuning. This prioritizes demonstrating robustness and generality of the experiment results.

### E.2.1 Hyperparameters Details of FM-IRL

We summarize the hyperparameters used by FM-IRL in Table 2. They cover the FM-enhanced discriminator, the FM vector field, distance-based reward, and training logistics.

Table 2: Details of hyperparameters in FM-IRL

| Name | Value | Meaning |
|---|---|---|
| fm_num_steps | 100 | FM time discretization steps (used for $t$ indexing in discriminator and for FM-based generation). |
| discrim_depth | 4 | Number of hidden layers in the FM discriminator's vector field $v_\theta$. |
| discrim_num_unit | 128 | Hidden width (units per layer) in $v_\theta$. |
| disc_lr | 1e-4 | Learning rate for the FM discriminator. |
| expert_loss_rate | 1.0 | Weight on expert branch loss term in discriminator training. |
| agent_loss_rate | -1.0 | Weight on agent branch loss term (negative encourages separation). |
| student_lr | 0.0001 | Learning rate of student policy. |
| reward_update_freq | 1 | Frequency to refresh rewards in the agent update loop (in updates). |
| state_norm | True | Whether to normalize states for the discriminator. |
| action_norm | False | Whether to normalize actions for the discriminator. |
| reward_norm | False | Whether to normalize rewards. |
| num_samples (MC) | 100 | Monte Carlo samples $S$ to estimate $Dist$ (expectation over $t$ and $x_0$), vectorized per-batch. |
| temperature | 0.1 | Scale factor on the averaged distance to stabilize magnitude in reward computation. |
| noise_scale | 0.5 | Standard deviation for $x_0$'s noise component used when forming $x_t$ during $Dist$ estimation. |

## F Additional Experiment Results

### F.1 Quantitative Results

We provides the quantitative results of our main experiments in Table 3.

### F.2 Hyperparameter Study

The hyperparameter $\beta$ in the policy regularization term (Eq. 11) plays a critical role in FM-IRL, as it directly regulates the strength of regularization and balances exploration against exploitation.

Table 3: Performance across six environments. For navigation and manipulation tasks, we report average success rate (Avg Suc. Rate); For locomotion tasks, we report average return (Avg Return). Values are presented as mean (±standard deviation).

| Algorithm | Navigation (Avg Suc. Rate) | | Manipulation (Avg Suc. Rate) | | Locomotion (Avg Return) | |
| --- | --- | --- | --- | --- | --- | --- |
| | (a) Ant-goal | (e) Maze2d | (b) Hand-rotate | (c) Fetch-pick | (d) Hopper | (f) Walker2d |
| DRAIL | 0.7142 (±0.0160) | 0.7780 (±0.0373) | 0.7775 (±0.2847) | 0.7052 (±0.3538) | 3182.60 (±85.25) | 3122.69 (±764.43) |
| GAIL | 0.6465 (±0.0542) | 0.6902 (±0.0826) | 0.9317 (±0.0541) | 0.2798 (±0.3316) | 2921.73 (±243.64) | 1698.25 (±411.42) |
| WAIL | 0.6127 (±0.0153) | 0.2978 (±0.0785) | 0.2370 (±0.3830) | 0.0000 (±0.0000) | 2609.28 (±814.05) | 1729.20 (±984.86) |
| VAIL | 0.7662 (±0.0365) | 0.6360 (±0.0382) | 0.5694 (±0.2960) | 0.8539 (±0.0551) | 2878.04 (±286.72) | 1156.52 (±221.67) |
| AIRL | 0.5467 (±0.0246) | 0.8239 (±0.0241) | 0.4595 (±0.1993) | 0.0000 (±0.0000) | 7.86 (±2.91) | -5.27 (±1.18) |
| DP | 0.8212 (±0.0135) | 0.5618 (±0.0268) | 0.9068 (±0.0136) | 0.8298 (±0.0127) | 1433.21 (±131.03) | 2204.41 (±226.28) |
| FP | **0.8334** (±0.0222) | 0.5420 (±0.0207) | 0.9032 (±0.0188) | 0.5460 (±0.0367) | 1950.41 (±170.32) | 2384.81 (±187.98) |
| FM-IRL (Ours) | 0.8225 (±0.0284) | **0.8731** (±0.0331) | **0.9794** (±0.0150) | **0.9984** (±0.0023) | **3358.95** (±72.31) | **4164.24** (±62.19) |

As shown in Figure 8, we conducted an ablation study on $\beta$ in the Fetch-pick environment, testing values of $\{0, 0.1, 0.5, 1, 2\}$ where $\beta = 0$ corresponds to no regularization (i.e. ablate regularization module). The results demonstrate that $\beta \in \{0.5, 1, 2\}$ all yield better convergence performance than $\beta = 0$, confirming the effectiveness of the regularization term. Specifically, $\beta = 2$ achieves the best performance among the tested values. Given that the optimal $\beta$ may vary across environments and the continuous parameter space admits infinitely many values, we select $\beta = 2$ for our main experiments for simplicity.

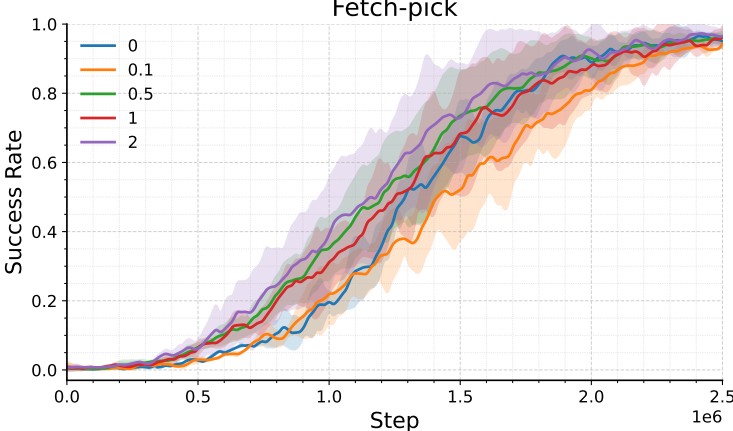

Figure 8: Performance of FM-IRL with different $\beta$ values in Fetch-pick environment.

## F.3 MORE GENERALIZATION STUDY

To further demonstrate the generalization ability of FM-IRL, we conduct more comprehensive experiments on the Navigation task Maze2d and the Locomotion task Walker2d. However, testing the agent's ability to generalize to unseen states requires different approaches for different task types. In Navigation tasks where the agent targets a specific location, applying noise to the goal position is not reasonable since it directly modifies the task objective itself, fundamentally changing what the agent is supposed to learn rather than testing generalization to state variations. Similarly, in Locomotion tasks where there are no explicit goal locations and the agent aims to maintain balance and move forward, the concept of goal noise is not applicable. Hence, for Navigation and Locomotion tasks, we measure the generalization ability to unseen out-of-distribution (OOD) states by decreasing the coverage or amount of expert data. Specifically, we establish 4 expert coverage settings in Maze2d: 25%, 50%, 75%, and 100%, where k% means that the expert demonstrations cover k% of all possible scenarios in this environment. In Walker2d, we directly control the number of $(s, a)$ transitions in the expert data, creating 4 settings: 2500, 2000, 1500, and 1000 transitions. We then compare the final performance of each method under each setting, as shown in Table 4. Similar conclusions as in Section 4.2 could be derived based on these results.

Table 4: Performance of different algorithms across different number of expert demonstrations. For Maze environment, we report average success rate (Avg Suc. Rate) w.r.t four levels of expert data coverage: 25%, 50%, 75%, 100%; For Walker2d environment, we report average return (Avg Return) w.r.t four levels of expert $(s, a)$ transitions in datasets.

| Algo. | Navigation: Maze (Avg Suc. Rate w.r.t expert coverage (%)) | | | | Locomotion: Walker2d (Avg Return w.r.t # demo transitions) | | | |
|---|---|---|---|---|---|---|---|---|
| | 25 | 50 | 75 | 100 | 1k | 1.5k | 2k | 2.5k |
| AIRL | $0.8143_{\pm 0.02}$ | $0.8528_{\pm 0.02}$ | $0.9017_{\pm 0.01}$ | $0.8729_{\pm 0.02}$ | $-7.23_{\pm 2.41}$ | $-2.51_{\pm 1.89}$ | $-5.94_{\pm 2.18}$ | $-4.62_{\pm 2.03}$ |
| FP | $0.5138_{\pm 0.02}$ | $0.6247_{\pm 0.02}$ | $0.7316_{\pm 0.04}$ | $0.7584_{\pm 0.04}$ | $1421.58_{\pm 118.74}$ | $1827.93_{\pm 136.52}$ | $2084.71_{\pm 161.38}$ | $2228.49_{\pm 174.26}$ |
| DP | $0.5267_{\pm 0.03}$ | $0.7419_{\pm 0.03}$ | $0.8436_{\pm 0.02}$ | $0.8917_{\pm 0.02}$ | $1572.84_{\pm 132.47}$ | $1896.25_{\pm 149.83}$ | $2173.52_{\pm 167.91}$ | $2468.37_{\pm 189.15}$ |
| DRAIL | $0.8059_{\pm 0.02}$ | $0.8516_{\pm 0.02}$ | $0.9128_{\pm 0.02}$ | $0.8642_{\pm 0.02}$ | $1458.76_{\pm 116.28}$ | $1691.84_{\pm 143.16}$ | $2947.39_{\pm 221.85}$ | $3066.72_{\pm 241.93}$ |
| GAIL | $0.6534_{\pm 0.04}$ | $0.6572_{\pm 0.04}$ | $0.8127_{\pm 0.03}$ | $0.8618_{\pm 0.03}$ | $903.47_{\pm 101.26}$ | $1158.65_{\pm 128.39}$ | $1275.84_{\pm 141.07}$ | $964.92_{\pm 108.75}$ |
| VAIL | $0.5926_{\pm 0.04}$ | $0.7438_{\pm 0.03}$ | $0.8015_{\pm 0.02}$ | $0.9024_{\pm 0.02}$ | $691.35_{\pm 90.18}$ | $679.47_{\pm 85.43}$ | $697.28_{\pm 91.86}$ | $967.51_{\pm 116.02}$ |
| WAIL | $0.3184_{\pm 0.05}$ | $0.1736_{\pm 0.05}$ | $0.1652_{\pm 0.05}$ | $0.5543_{\pm 0.04}$ | $901.28_{\pm 161.35}$ | $922.16_{\pm 173.94}$ | $996.57_{\pm 184.03}$ | $2457.85_{\pm 322.16}$ |
| **FM-IRL (ours)** | $\mathbf{0.8273}_{\pm 0.01}$ | $\mathbf{0.9231}_{\pm 0.01}$ | $\mathbf{0.9426}_{\pm 0.01}$ | $\mathbf{0.9718}_{\pm 0.01}$ | $\mathbf{3026.81}_{\pm 70.53}$ | $\mathbf{3635.42}_{\pm 84.91}$ | $\mathbf{3812.68}_{\pm 94.07}$ | $\mathbf{4368.93}_{\pm 108.92}$ |

### F.4 SUPPLEMENTARY EXPERIMENTS FOR FAIR-IRL-REWARDS

We present the experiment results when all four methods adopts the same reward-the learned reward function via FM-enhanced discriminator. Then, the only difference lies in the policy head: FM-IRL adopts simple MLP-based gaussian policy, while the other three methods adopts Flow Matching Generative Policy (namely, the action is generated from a random noise $\mathcal{N}(0, I)$ via an ODE solver).

The results in Table 5 shows that the similar pattern as in Section 4.4-three methods to online update FM policy fail to learn a feasible policy no matter the reward function is true reward or learned reward. From these observations, we can conclude that with a sophisticated discriminator to learn a meaningful reward, the MLP gaussian policy is **sufficient** to learn a policy which is high-performing and robust during deployment. In contrast, FM-like generative policy is hard to optimize and will potentially cause extreme instability, resulting in the poor performance regardless of the reward form. These insights also align and consolidate the motivation of the design of FM-IRL.

Table 5: Performance comparison between FM-IRL and FM-A2C, FM-PPO and FPO in Maze environment with learned reward function

| Algorithm | Avg Success Rate |
|---|---|
| FM-A2C | $0.0950_{\pm 0.04}$ |
| FM-PPO | $0.1250_{\pm 0.08}$ |
| FPO | $0.0533_{\pm 0.01}$ |
| FM-IRL | $\mathbf{0.8750}_{\pm 0.03}$ |

## G BASELINE DETAILS

In this section, we summarize the methodological foundations of all baselines used in our comparisons. When possible, we present their core objectives in mathematical form for clarity and reproducibility.

**Diffusion Policy (DP) (Chi et al., 2024).** DP models the conditional action distribution $\pi(a \mid s)$ via a conditional diffusion generative model. Let $\{q_t\}_{t=0}^{T}$ denote the forward noising process that gradually perturbs clean actions $a_0 \sim \pi_E(\cdot \mid s)$ into noisy variables $a_t$:

$$q(a_t \mid a_{t-1}) = \mathcal{N}\big(\sqrt{1 - \beta_t}\, a_{t-1}, \beta_t I\big), \quad t = 1, \ldots, T. \qquad (34)$$

A reverse-time parameterization $p_\theta$ denoises step-by-step conditioned on $s$:

$$p_\theta(a_{t-1} \mid a_t, s) = \mathcal{N}\big(\mu_\theta(a_t, s, t), \Sigma_\theta(a_t, s, t)\big). \qquad (35)$$

Training minimizes the denoising score-matching loss (often in the $\epsilon$-prediction parameterization):

$$\mathcal{L}_{\mathrm{DP}}(\theta) = \mathbb{E}_{t, (s, a_0) \sim \mathcal{D}_E, \epsilon \sim \mathcal{N}(0, I)} \big\| \epsilon_\theta(a_t, s, t) - \epsilon \big\|^2, \qquad (36)$$

where $a_t = \sqrt{\bar{\alpha}_t}\, a_0 + \sqrt{1 - \bar{\alpha}_t}\, \epsilon$ and $\bar{\alpha}_t = \prod_{i=1}^{t}(1 - \beta_i)$. Inference samples $a_T \sim \mathcal{N}(0, I)$ and iteratively applies $p_\theta$ to obtain $a_0$. DP is a supervise learning approach to clone the expert behavior.

**Flow-Matching Policy (FP) (Zhang et al., 2025a).** FP models $\pi(a \mid s)$ via conditional flow matching. Let $a_t$ follow an ODE driven by a conditional vector field $v_\theta$:

$$\frac{da_t}{dt} = v_\theta(a_t, s, t), \quad a_0 \sim \mathcal{N}(0, I), \quad a_1 \equiv a. \tag{37}$$

With a predefined conditional path $p_t(a \mid s, a_1)$ and teacher velocity $u_t(a_t \mid s, a_1)$, FP minimizes the FM regression:

$$\mathcal{L}_{\text{FP}}(\theta) = \mathbb{E}_{t \sim \mathcal{U}[0,1],\,(s,a_1) \sim \mathcal{D}_E,\, a_t \sim p_t(\cdot \mid s, a_1)} \big\| v_\theta(a_t, s, t) - u_t(a_t \mid s, a_1) \big\|^2. \tag{38}$$

At test time, $a_1$ is obtained by numerically integrating the ODE from $a_0 \sim \mathcal{N}(0, I)$. FP is a supervise learning approach to clone the expert behavior.

**GAIL (Ho & Ermon, 2016).** GAIL frames imitation as matching occupancy measures via an adversarial game between policy $\pi_\phi$ and discriminator $D_\psi$:

$$\min_\psi \max_\phi \; \mathbb{E}_{(s,a) \sim \rho_{\pi_\phi}} \big[ \log D_\psi(s, a) \big] + \mathbb{E}_{(s,a) \sim \rho_E} \big[ \log(1 - D_\psi(s, a)) \big] - \lambda \, \mathcal{H}(\pi_\phi), \tag{39}$$

where $\mathcal{H}$ is policy entropy regularization, $\rho_{\pi_\phi}$ is the state-action visitation distribution under $\pi_\phi$. The shaped reward for RL is $r(s, a) = -\log(1 - D_\psi(s, a))$.

**VAIL (Peng et al., 2018).** VAIL augments GAIL with an information bottleneck on the discriminator via a variational latent $z$ to reduce overfitting and improve robustness:

$$D_\psi(s, a) = \sigma\big(f_\psi(s, a, z)\big), \quad z \sim q_\psi(z \mid s, a), \tag{40}$$

and adds a KL constraint to enforce an information bottleneck:

$$\mathcal{L}_{\text{IB}}(\psi) = \beta \, \mathbb{E}_{(s,a)} \big[ \text{KL}\big(q_\psi(z \mid s, a) \,\|\, p(z)\big) \big], \tag{41}$$

leading to the min-max:

$$\min_\psi \max_\phi \; \mathbb{E}_{\rho_{\pi_\phi}} \big[ \log D_\psi(s, a) \big] + \mathbb{E}_{\rho_E} \big[ \log(1 - D_\psi(s, a)) \big] + \mathcal{L}_{\text{IB}}(\psi) - \lambda \mathcal{H}(\pi_\phi). \tag{42}$$

**WAIL (Xiao et al., 2019).** Wasserstein Adversarial Imitation Learning replaces the JS divergence in GAIL with the 1-Wasserstein distance using a 1-Lipschitz critic $f_\psi$:

$$\max_{\psi \in \text{Lip}(1)} \; \mathbb{E}_{\rho_{\pi_\phi}} \big[ f_\psi(s, a) \big] - \mathbb{E}_{\rho_E} \big[ f_\psi(s, a) \big], \tag{43}$$

with gradient penalty enforcing Lipschitzness:

$$\mathcal{L}_{\text{GP}}(\psi) = \lambda_{\text{gp}} \, \mathbb{E}_{\hat{x}} \big( \|\nabla_{\hat{x}} f_\psi(\hat{x})\|_2 - 1 \big)^2, \quad \hat{x} = \epsilon x_E + (1 - \epsilon) x_\pi. \tag{44}$$

The policy is trained with reward $r(s, a) = f_\psi(s, a)$.

**AIRL (Fu et al., 2017).** AIRL decomposes the discriminator to recover a reward function invariant to dynamics:

$$D_\psi(s, a, s') = \frac{\exp\big(f_\psi(s, a, s')\big)}{\exp\big(f_\psi(s, a, s')\big) + \pi_\phi(a \mid s)}, \quad f_\psi(s, a, s') = g_\psi(s, a) + \gamma h_\psi(s') - h_\psi(s), \tag{45}$$

where $g_\psi$ approximates the reward and $h_\psi$ the shaping potential. The implied reward for policy optimization is $r_\psi(s, a) = g_\psi(s, a)$.

**DRAIL (Lai et al., 2024).** DRAIL replaces the GAIL discriminator with a conditional diffusion model trained as a binary classifier via single-step denoising losses. For a state-action pair $(s, a)$ and condition $c \in \{c^+, c^-\}$ (expert vs. agent), define the class-conditional diffusion loss

$$\mathcal{L}_{\text{diff}}(s, a, c) = \mathbb{E}_{t, \epsilon} \, \|\epsilon_\phi(s, a, \epsilon, t \mid c) - \epsilon\|_2^2, \tag{46}$$

approximated with a single sampled $(t, \epsilon)$. Let $\mathcal{L}_{\text{diff}}^\pm(s, a) \equiv \mathcal{L}_{\text{diff}}(s, a, c^\pm)$. The diffusion discriminative classifier is

$$D_\phi(s, a) = \frac{e^{-\mathcal{L}_{\text{diff}}^+(s,a)}}{e^{-\mathcal{L}_{\text{diff}}^+(s,a)} + e^{-\mathcal{L}_{\text{diff}}^-(s,a)}} = \sigma\big(\mathcal{L}_{\text{diff}}^-(s, a) - \mathcal{L}_{\text{diff}}^+(s, a)\big). \tag{47}$$

Train $D_\phi$ with BCE:

$$\mathcal{L}_D = \mathbb{E}_{\tau_E}[-\log D_\phi(s, a)] + \mathbb{E}_{\tau_i}[-\log(1 - D_\phi(s, a))]. \tag{48}$$

Policy optimization uses the adversarial (logit) reward

$$r_\phi(s, a) = \log D_\phi(s, a) - \log(1 - D_\phi(s, a)), \tag{49}$$

and any RL optimizer (e.g., PPO). This design avoids costly full diffusion sampling, yields a bounded, smooth "realness" signal, and aligns with GAIL's occupancy-matching objective via a class-conditioned diffusion discriminator.

**Implementation Notes.** All IRL baselines are trained with on-policy or off-policy RL updates using the shaped rewards defined above. Supervised baselines (DP and FP) are trained purely on $\mathcal{D}_E$ without online interaction, hence their evaluation curves are horizontal as they do not improve with additional environment steps.

### G.1 Comparison between FM-IRL (Ours) and DRAIL

Compared to DRAIL, FM-IRL introduces several key improvements: 1) Enhanced Discriminator efficiency via Flow Matching: Our method employs an FM-based discriminator grounded in Optimal Transport, which requires fewer time discretization steps to generate high-quality state-action samples. In contrast, DRAIL relies on a diffusion-based discriminator that demands multiple sampling steps to achieve comparable accuracy. 2) Flexible Probability Paths: The FM-enhanced discriminator in our framework supports a broader family of probability paths from noise to target state-action pairs, benefiting from the expressiveness of FM in adversarial imitation learning. DRAIL, however, is constrained to pre-defined diffusion processes, limiting the flexibility and representational capacity of discriminator compared to FM. 3) Integrated Regularization: FM-IRL leverages the same FM model as a generator to regularize the policy at minimal computational cost. This is particularly important in practice: although FM or diffusion-based discriminators exhibit stronger expressiveness on in-distribution data, they may generalize poorly to unseen samples. Thus, the regularization mechanism is essential to balance exploration and exploitation.

## H    Summary of Related Work

This section presents a comparative table summarizing the key characteristics of existing related works (Table 6). It is important to highlight that although all the listed works integrate generative models with decision-making algorithms, they belong to distinct settings.

Specifically, DQL, IDQL, and FQL fall under offline reinforcement learning, where policies are trained from a static dataset containing states, actions, and rewards. In contrast, DPPO and ReinFlow investigate the fine-tuning of pre-trained flow matching models in online environments. Meanwhile, DIPO operates within an online reinforcement learning setting, training a critic network using the true reward function provided by the environment.

Our approach, however, follows an imitation learning setting (namely, we have access to expert demonstrations, but not access to true reward of environment), which fundamentally differs from the methods mentioned above. Notably, Diffusion Policy, Flow Matching Policy, the GAIL family, as well as DRAIL, share the same setting as ours, and are therefore selected as baselines for comparison.

Table 6: Comparison of policy learning frameworks across key dimensions

| Method | Online Expl. | Dist. Match | Stable Grad. | No Post-train | Efficient Inf. | Reg. |
|---|---|---|---|---|---|---|
| Diffusion Policy | | ✓ | ✓ | ✓ | ✓ | |
| Flow Matching Policy | | ✓ | ✓ | ✓ | ✓ | |
| DQL / IDQL | | ✓ | | ✓ | | |
| FQL | | ✓ | ✓ | ✓ | ✓ | ✓ |
| DIPO | ✓ | ✓ | ✓ | ✓ | | |
| DPPO | ✓ | ✓ | | | | |
| ReinFlow | ✓ | ✓ | | | ✓ | |
| GAIL Family | ✓ | | ✓ | ✓ | ✓ | |
| DiffAIL / DRAIL | ✓ | ✓ | ✓ | ✓ | ✓ | |
| **FM-IRL (Ours)** | ✓ | ✓ | ✓ | ✓ | ✓ | ✓ |

Note: This table summarizes which capabilities are *explicitly supported* by each framework.
Key dimensions:
- *Online Expl.*: Supports active exploration through environmental interaction
- *Dist. Match*: Captures complex multi-modal distributions
- *Stable Grad.*: Avoids backpropagation instability
- *No Post-train*: Learns from scratch without fine-tuning
- *Efficient Inf.*: Enables low-latency policy execution
- *Reg.*: Uses behavior regularization

## I   STATEMENT OF LLM USAGE

This paper employed LLM (specifically, GPT-5) to enhance the language quality of the writing. All content originated from the author, and the LLM was used solely to refine grammar, improve vocabulary usage, and enhance sentence-level coherence, without altering the original meaning.

