# OpenReview forum: "FM-IRL: Flow-Matching for Reward Modeling and Policy Regularization in Reinforcement Learning"
_ICLR.cc/2026/Conference — Submitted to ICLR 2026_

### Official Review · Reviewer_N3MV · 2025-10-15

**Soundness:** 2
**Presentation:** 3
**Contribution:** 2
**Rating:** 2
**Confidence:** 3

**Summary:**

This paper aims to incorporate the benefits of online exploration into flow matching training by building on an inverse reinforcement learning framework. First, a teacher flow-matching model is trained on a static offline dataset. To enhance the expressiveness of the reward model, the teacher model’s flow-matching loss on agent-generated trajectories is used to measure the discrepancy between the agent distribution and the expert distribution. To avoid the instability of updating flow-matching policies through backpropagation through time or policy gradients, the method adopts a distillation objective that jointly integrates the rewards and teacher behavior into a simpler student policy. Experimental results show that, with additional online interactions, the proposed approach can mitigate the potential sub-optimality of offline datasets and outperform standard behavior cloning models trained solely on static data.

**Strengths:**

1. The paper is clearly written and well structured.

2. Extending the capabilities of flow-matching or diffusion policies on out-of-distribution area using online interatctions is an important research direction, since distributional shift is hard to be addressed by simply scaling static offline data. It is necessary to using additional online interatctions of model rollouts to mitigate some corner cases.

3. Improving the RL training stability for flow-matching or diffusion policies is important. Most current methods like directly maximizing differentiable rewards or policy gradients can be unstable.

**Weaknesses:**

1. `Lack of Novelty`.

This paper appears to be a straightforward combination of three existing ideas: using diffusion losses as rewards [1], applying distillation methods to optimize flow-matching policies via reinforcement learning [2], and inverse reinforcement learning [3]. Although the authors claim novelty in being the first to use flow-matching loss as rewards, flow matching and diffusion models are essentially two sides of the same coin. Therefore, this contribution does not strike me as genuinely novel.

Furthermore, the idea of using reinforcement learning to optimize a distilled, simpler policy is already well studied [2][4][5]. As a result, using distillation to enhance the stability of flow-matching RL training does not appear novel either. Taken together, these factors make the paper resemble a naïve “A + B + C” combination without a clearly original contribution.

2. `Weak Motivation`.

If the authors had provided strong motivation for why the “A, B, C” components should be integrated, I could have acknowledged the contribution despite the limited novelty. Unfortunately, the current manuscript fails to do so.

The authors identify the potential suboptimality of offline datasets in traditional flow-matching training as the core challenge, and aims to introduce additional online interactions to address this. However, the final objectives still primarily fit the static offline dataset. For example, rewards are higher when the agent’s behavior resembles that of the offline dataset, which only encourages in-distribution behavior. In addition, to improve training stability and mitigate inaccurate reward estimation on out-of-distribution areas, the authors introduce a distillation loss that encourages the student policy to mimic the teacher flow. This, again, is essentially behavior cloning.

In my view, the proposed method reformulates the “mimic offline dataset” objective into “A + B + C jointly mimic the offline dataset.” This reformulation does not fundamentally address the challenge that the authors themselves emphasize. For this reason, I would recommend rejection.

3. `Marginal Improvements`.

The experimental results show only marginal improvements over baseline models, which further limits the paper’s contribution.

[1] Diffusion-reward adversarial imitation learning. 2024

[2] Flow q-learning. 2025

[3] Generative adversarial imitation learning. 2016

[4] Score Regularized Policy Optimization through Diffusion Behavior. 2024

[5] Diffusion Policies creating a Trust Region for Offline Reinforcement Learning. 2024

**Questions:**

Please see weaknesses for details.

---

> ### Author Response · Authors · 2025-11-18
> **Author Response (1)**
>
> ## Thanks for the valuable questions, and below is our detailed response.
>
> ---
> **Q1.** FM-IRL appears to be a straightforward combination of diffusion loss, distillation and IRL, making the contribution not novel.
>
> **A1.** We respectfully disagree with the characterization of our work as "naive A+B+C combination," we acknowledge that our manuscript may not have sufficiently articulated the fundamental conceptual shift and problem formulation novelty that distinguishes FM-IRL from prior work. Let us address each concern systematically.
>
> **Our design follows a rigorous motivation chain, not component combination.** Our core contribution is an architectural innovation, a teacher-student framework, that addresses a fundamental limitation: traditional FM policies cannot perform online updates and lack exploration capabilities, resulting in the lack of generalization abilities (as proved in our experiments, Section 4.2, 4.3). This motivation is entirely different from DRAIL's. Since FM policies can only fit offline datasets and cannot update online, we leverage FM's strength by having it fit the expert dataset while disentangling exploration to a separate student policy that explores online under FM guidance. This corresponds to our "infusion" mechanism in the paper.
>
> However, this design introduces a new challenge: FM's powerful ability to fit complex distributions means the reward signal may overfit to expert data. When exploration reaches OOD states, reward estimation becomes unreliable. To address this, we introduce a regularization term that balances exploration and exploitation, **reusing the FM model** as part of our novel teacher-student architecture. Each component directly solves problems arising from the preceding design choice.
>
> **Distinction from FQL's distillation:** The reviewer mentions FQL uses similar distillation ideas to our regularization term. However, FQL's distillation and our infusion operation are fundamentally different: (1) **Problem setting**: FQL operates in offline RL with direct access to rewards in the dataset. Our setting is imitation learning without reward access. (2) **Technical details**: FQL conditions on state $s$ and distills action distribution from a multi-step FM to a single-step FM. We directly infuse the joint $(s,a)$ distribution (reasoning explained in our paper) from FM into an MLP-based student policy. These are fundamentally different operations serving different purposes in different problem settings.
>
> ---
> **Q2.** Does online interaction truly provide benefits when rewards come solely from offline data? Since the reward function is learned from an offline dataset, it can only encourage in-distribution behavior, and regularization further restricts exploration---making the framework essentially mimic the offline dataset through joint optimization.
>
> **A2.** This is a quite insightful question. We added the discussion about this insight in Appendix B.2 (l.853), and provide the content here for your convenience:
>
> Actually, while the reward model is trained on offline data, it learns a **generalizable reward function** capable of evaluating arbitrary $(s,a)$ pairs, including those never observed. Consider RLHF: it learns from a fixed human feedback dataset yet enables models to exhibit emergent generalization capabilities on unseen samples. The reward function is a **learned abstraction** that captures the concept of **"expert-like behavior"** rather than memorizing specific $(s,a)$ pairs.
>
> Characterizing our approach as "mimicking the offline dataset" is an oversimplification. We are actually **learning the expert's task objective**, which is fundamentally different. Behavioral cloning methods (e.g., Diffusion Policy, Flow Matching Policy) directly memorize $(s,a)$ mappings with poor generalization. Naive IRL methods like GAIL match state-action occupancy measures point-wise, still suffering from limited generalization. FM-IRL learns the **distributional structure of expert data**, enabling the model to understand the expert's objective—what makes expert behavior successful—and infer whether other $(s,a)$ pairs align with this objective.
>
> Regarding regularization: it does not exacerbate distribution-matching. Rather, regularization constrains the exploration space, preventing the agent from venturing into highly out-of-distribution states where reward estimates become unreliable.
>
> **Clarification:** Perhaps we overstated "online learning." More precisely, our contribution is enabling better generalization through guided online interaction—using offline-learned rewards to facilitate principled exploration beyond the demonstration dataset.
>
> ---
> **Q3.** The experiment results in Navigation tasks show marginal improvement.
>
> **A3.** This is a critical observation and aligns with the design principle and strength of FM-IRL. Please refer to *Author Response (2)* for our explanation due to the character limit.
>
> ---
> ## We sincerely hope your concerns are fully addressed.

---

> > ### Comment · Reviewer_N3MV · 2025-11-25
> > **Follow ups**
> >
> > Q2. it learns a generalizable reward function ....
> >
> > The authors use RLHF as an example to argue that learned reward models can generalize to OOD regions. However, I do not believe this is an adequate analogy. RLHF reward models are trained to contrast positive and negative preference pairs. The objective is not to encourage in-distribution behavior, but rather to prompt positive behaviors while avoiding behaviors labeled as negative. In contrast, the reward objective in this paper explicitly contrasts in-distribution expert data with OOD data. Under such a setup, it is unclear how the method can extract an abstract notion of “expert-like behavior” that would reliably generalize to OOD regions.
> >
> > More broadly, OOD generalization is notoriously difficult to explain in deep neural networks and is largely determined by the coverage of the training data distribution. From a methodological perspective, I therefore remain unconvinced that the learned reward in this paper is genuinely generalizing to OOD areas. Instead, it appears to primarily encourage staying close to in-distribution expert behavior.

---

> > > ### Author Response · Authors · 2025-11-25
> > >
> > > Thanks for the follow-up.
> > >
> > > We apologize for the confusion caused. Actually, we are referring to **RL post-training of LLM** as our analogy rather than RLHF specifically. Consider *math* tasks where state-of-the-art techniques utilize RL to train the LLM's reasoning capabilities based on a fixed dataset of math questions and answers. Please note that the reward is only available for the questions in the fixed training data. Yet, the model is able to generalize to more complex and longer OOD reasoning problems [1]. How?
> > >
> > > - 1. RL is not just training the model to memorize the answers. Instead, through reflection based on the reward signal derived from the training data, it learns a generalizable reasoning strategy. It backpropagates the logical correctness (reward from static data) to every step of the Chain-of-Thought (CoT), thereby cultivating a capability for 'logical self-consistency'. This is very similar to our previous claim that we ''learn expert-like behavior'' or ''learn the objective of the expert''. Although our reward originates from static data, it extracts the task objective (i.e., the correct flow of behavior), just as Math RL extracts the universal logical process from static data. Both of these methods utilize RL to elevate the model from the level of ''mimic'' to ''understanding''. The success of LLM post-training via RL forms the motivation of our method.
> > >
> > > - 2. Furthermore, traditional discriminators (like GAIL) in OOD regions only output a probability $P \approx 0$ (telling you, 'you are wrong'). FM-IRL, however, learns a **Vector Field**. Even in regions slightly deviating from the expert distribution (**Local OOD**), the Flow field is still able to provide some encouraging reward. It contains directional information, which does not just tell you ''you are wrong,'' but also tells you ''which direction to move to return to the Expert objective.'' This is the source of our generalization.
> > >
> > > - 3. Moreover, regarding the distillation loss you mentioned that encourages the student policy to mimic the teacher flow: State-of-the-art methods for RL post-training of LLMs always add a regularization term to prevent the model's parameters from deviating too far from a **reference** model. Nevertheless, this does not prevent them from generalizing to OOD samples [1]. We borrow this inspiration by treating our **reference model** as the FM teacher, as the FM teacher is as capable as a foundation model in our Navigation/Locomotion/Manipulation context.
> > >
> > > [1] DeepSeek-R1: Incentivizing Reasoning Capability in LLMs via Reinforcement Learning, 2025
> > >
> > > ---
> > > ## We hope your concerns are fully addressed, and would express our gratitude for your insight to refine our paper.

---

> > > > ### Comment · Reviewer_N3MV · 2025-11-27
> > > > **Follow ups**
> > > >
> > > > Thanks for the additional replies. However, several concerns remain unresolved.
> > > >
> > > > 1. The source of generalization for the learned rewards
> > > >
> > > > The authors argue that the velocity of the flow model (FM) not only reflects the expert likelihood but also indicates the direction toward the expert objective. However, this idea has already been explored in prior work such as DRAIL [1] and other approaches that use FM/Diffusion-based signals as rewards. Although the authors claim that the motivation in this paper differs from that of DRAIL, the application scenarios and resulting high-level algorithmic structures appear highly similar.
> > > >
> > > > 2. Is it merely a regularization technique, or does it prevent OOD generalization?
> > > >
> > > > This largely depends on the regularization strength controlled by $\beta$. In the offline RL literature, $\beta$ is typically introduced to prevent policies from exploiting OOD samples. However, simply adding a regularization term cannot fully eliminate OOD samples, which inevitably leads to some degree of OOD generalization. But, using a large $\beta$ can still significantly hinder OOD exploration. So, more experiments can strength this claim.
> > > >
> > > > [1] Diffusion-Reward Adversarial Imitation Learning, 2024.

---

> ### Author Response · Authors · 2025-11-21
> **Author Response (2)**
>
> **Q3.** The experiment results in some environments (like Navigation tasks) show marginal improvement.
>
> **A3.** This observation exactly aligns with the strength and motivation of FM-IRL. Please note that the advantage of our FM-enhanced discriminator lies in capturing multi-modal expert distributions during training. In navigation tasks, the objective is to move towards a target position. The success trajectory is often unimodal and is fully covered by expert data, obviating the need of excessive exploration and multi-modal training. Therefore, FM-IRL shows marginal improvement but in such relatively "uni-modal tasks”, FM-IRL roughly degraded to the strongest baseline, preserving its lower-bound performance. In Manipulation tasks, the goal is not anymore “moving to a target place” but to try to manipulate an object, resulting in more complexity, flexibility and multi-modality in expert data. In such tasks, FM-IRL **significantly outperforms baselines**, validating our motivation. Meanwhile, our method introduces much-lower variance in tasks like Hopper, Fetch-pick and **significant performance improvement** in Walker2d, validating the effectiveness of regularization term to stabilize the training. We have added this discussion in Appendix B.1 (l.788).

---

> ### Author Response · Authors · 2025-11-27
>
> ## We thank the reviewer for the efficient follow-up. We hereby address your further concerns.
>
> ---
> **Q1.** The idea of FM to output a direction has been explored in prior work like DRAIL.
>
> **A1.** Actually, we have already acknowledged in our paper that our work is partly inspired by prior work like DRAIL (e.g., see l.233). Moreover, we argue that drawing inspiration from prior work is not mimicking prior work:
> - Our motivation totally differs from DRAIL (as you already mentioned).
>
> - Although inspired, we identified the potential limitation of DRAIL and addressed them by proposing dual-purpose FM model and teacher-student architecture. We have already provided the detailed discussion in our manuscript about the technical improvement between DRAIL and our method (please see l. 1372).
>
> - Aligning with these insights, our method is proved to significantly outperforms DRAIL in more than half of the benchmarks in our experiments.
>
> These makes our work not only inspired but also one step further than DRAIL, contributing to the family of applying generative model to enhance IRL.
>
> **Q2.** Is it merely a regularization technique, or does it prevent OOD generalization? More experiments can strength this claim.
>
> **A2.** Thanks for this insightful question. We fully agree that using a small $\beta$ could inevitably lead to OOD generation but a large $\beta$ can still significantly hinder OOD exploration. We have already conducted the hyperparameter study about $beta$ in Appendix F.2 and kindly refer you to our paper (l.1210) for details. There, we tested different value of $\beta$ and monitor its impact on convergence speed and ultimate performance. However, the best value of $\beta$ differs from environment to environment, so we selected the best $\beta$ value in Fetch-pick for all environments in our experiment without further hyperparameter-tuning (though such behavior could potentially further improve the performance of our method).
>
>
> ---
> ## Thanks again for this reviewer's prompt follow-up, and we are willing to address your further concerns in our earliest effort, since the deadline of rebuttal session is approaching.

---

> > ### Comment · Reviewer_N3MV · 2025-11-27
> > **Thanks for the rebuttal**
> >
> > Most of my concerns are resolved and I am happy to raise my score to a 6.

---

> ### Author Response · Authors · 2025-11-27
> **Thanks for re-considering the score**
>
> The authors are glad to see that most of your concerns are fully addressed, and we'd again express our gratitude for your time and effort to review our paper. Your insight would be very helpful to improve our paper.

---

### Official Review · Reviewer_XrqD · 2025-10-30

**Soundness:** 3
**Presentation:** 3
**Contribution:** 3
**Rating:** 8
**Confidence:** 3

**Summary:**

This paper proposes an offline imitation learning framework in which a student policy learns from a reward model based on Flow Matching (FM). The authors begin by noting that the absence of an online FM policy learning mechanism limits the policy's generalization capability. The point of this paper is not to train a FM policy. Inspired by adversarial inverse reinforcement learning, this work leverage FM to develop an enhanced discriminator. A student policy is implemented as a simple MLP. The FM-based discriminator is trained to fit expert data while distinguishing it from the behavior generated by the student policy.

The authors also observe that while several prior attempts have integrated online RL with diffusion models, these methods often suffer from training instability. I would recommend that the authors also discuss the relevant work DACER [1] in this context, but I generally agree with this statement based on my own experience. Furthermore, the paper elaborates on the inherent challenges of training FM policies online. Its key contribution lies in leveraging a powerful generative model to "infuse" knowledge into a simple policy, while the simple policy also learns online to prevent overfitting.

Overall, this paper is interesting to me. However, I still have some concerns about the comprehensiveness of the technical details, which lower my confidence.

[1] Wang, Yinuo, et al. "Diffusion actor-critic with entropy regulator." *Advances in Neural Information Processing Systems* 37 (2024): 54183-54204.

**Strengths:**

This paper presents a well-motivated and novel approach, supported by a clear and logical structure. The proposed method demonstrates significant performance improvements over baselines. The authors also provide comprehensive discussion in the appendix, including answers to some possible questions, which greatly aids in understanding the methodological rationale.

**Weaknesses:**

I am not clear about whether the framework is easy to implement effectively or it requires tricks and careful hyperparameter tuning. Also, I am not sure about how the authors made the comparison fair (e.g., using common hyperparameters or network architectures, or fine-tuning each algorithm one-by-one). I noticed that the authors claim that the code will be made open-source, but I would appreciate some explicit discussion about such details.

**Questions:**

1. Is there a learned value function or advantage estimation for the student MLP policy? Regarding the student policy loss (Equation (11)), the first term is the expected return. Does this mean that the student policy learning is identical to REINFORCE with adversarial rewards when $\beta = 0$?
2. How is the MLP student policy implemented? For example, PPO usually outputs a mean vector and uses a fixed scale to represent a Gaussian distribution, while SAC typically outputs both a mean vector and a per-dimension scale vector. There is also a policy class called amortized actors [2,3,4] which, though structurally an MLP, can express a multi-modal decision distribution.
3. The teacher FM can represent a multi-modal data distribution, but the student policy probably cannot (if it is Gaussian). In cases where the expert data is highly multi-modal (for example, the scenario discussed in Figure 1 of the DQL paper (Wang et al., 2022)), would the "infusing" encounters challenges?
4. Are there any tricks or hyperparameters not covered in the appendix? For example, only disc_lr is listed in Table 2. Is this a global learning rate that also applies to the student policy? What is the detailed network architecture of FM teacher?
5. How did you made the comparison with baselines fair?
6. What is $p_\theta$ and $T$ in equation (1)? They don't seem to have been explained.

[2] Haarnoja, Tuomas, et al. "Reinforcement learning with deep energy-based policies." *International conference on machine learning*. PMLR, 2017.

[3] Messaoud, Safa, et al. "S$^ 2$AC: Energy-Based Reinforcement Learning with Stein Soft Actor Critic." *12th International Conference on Learning Representations, ICLR 2024*. 2024.

[4] Wang, Ziqi et al. "Learning Intractable Multimodal Policies with Reparameterization and Diversity Regularization." *Advances in Neural Information Processing Systems*. 2025.

---

> ### Author Response · Authors · 2025-11-18
>
> ## We appreciate the valuable questions, and below is our detailed response.
> ---
> **Q1.** I am not clear about whether the framework is easy to implement effectively or it requires tricks and careful hyperparameter tuning.
>
> **A1.** Sorry for confusion. Actually, our method is **easy to implement**, and we have added the discussion about implementation in Appendix E.1.3 (l. 1118) and E.2 (l.1146).
>
> We provide the concise version of the discussion for your convenience: A notable strength of FM-IRL lies in its algorithmic elegance: its reward model is designed entirely within the GAIL framework while addressing the limitations of traditional GAIL-family methods. This architectural choice ensures that FM-IRL can be implemented with minimal modifications to standard GAIL codebases. **Practitioners only need to implement the computation details of $D$ while maintaining interface compatibility with GAIL’s discriminator API**; the rest of the training pipeline remains identical. There are no complex tricks and careful hyperparameter tuning.
>
> ---
> **Q2.** How does the authors make the comparison fair?
>
> **A2.** Thanks for the valuable question. We have added the discussion about fair comparison in Appendix E.2 (l. 1147).
>
> We provide the the discussion for your convenience: To ensure fair comparison, we adopt a practical two-tier strategy for hyperparameter configuration. For **shared** components, like policy and critic networks, learning rates, batch sizes, and discount factors, all methods use **identical settings**. This ensures that performance differences reflect genuine algorithmic improvements rather than implementation advantages. For FM-IRL's unique components (FM-enhanced discriminator), we deliberately choose **one straightforward set of hyperparameters** and fix them across all tasks without per-task tuning. This prioritizes demonstrating robustness and generality of the experiment results. For baselines, we use the hyperparameter settings in their **original repository**.
>
> ---
> **Q3.** Does student MLP policy have a learned value function? Is it the same with REINFORCE with adversarial rewards when $\beta=0$?
>
> **A3.** We have added these discussion in Appendix E.1.4 (l.1134) and l.1127-1133.
>
> We provide a concise version of these discussion for your convenience: Yes, when the regularization weight $\beta=0$, the training of the student MLP policy **strictly follows the standard paradigm of PPO**, including learning a value function/advantage function (critic network) based on reward model, modeling the policy as MLP with Gaussian head, and performing trust-region policy optimization based on PPO-objective. **This also makes the implementation of our method easier**, since the researcher only need to follow the PPO codebase and just modify the reward function by adversarial rewards.
>
> ---
> **Q4.** How is PPO policy implemented? What is the output of the policy network?
>
> **A4.** The PPO in our implementation outputs a mean vector and a per-dimension scale vector. The policy is a diagonal Gaussian with a **state-dependent learnable mean $\mu$ and single learnable log-std vector $\Sigma$** of size *action\_dim*. $\Sigma$ is a global **state-independent** parameter. It is not a fixed constant and not state-dependent. Using a single global learnable log‑std vector for PPO’s diagonal Gaussian policy improves stability and simplicity: it decouples exploration scale from state, reducing gradient variance, typically yielding more stable, reproducible training from our experiment experience.
>
> ---
> **Q5.** Would the "infusing" be challenging since the deployment policy is uni-modal?
>
> **A5.** This is a very insightful question. We kindly refer you to our response to Q1 of reviewer 1s8J, or see our discussion in main body of our paper (l.467) for details.
>
> ---
> **Q6.** Only disc\_lr is listed in Table 2. Is this a global learning rate that also applies to the student policy? What is the detailed network architecture of FM teacher?
>
> **A6.** Sorry for confusion. Actually, this is only the learning rate for discriminator and we added the global learning rate of student policy in Table2 (l.1168). On the other hand, we use the hyperparameters of the original implementation of baselines [1] and will open source all the hyperparameter details when our paper is published.
>
> The part of FM teacher being parametrized is actually its velocity field $v_\theta$, and we simply use the MLP architecture to model the $v_\theta$ following standard FM paradigm.
>
> ---
> **Q7.** Part of the notations in equation 1 are not explained.
>
> **A7.** Sorry for confusion. $p_\theta$ means the **probability density function** corresponding to specific vector field parametrized by $\theta$, $T$ means the number of **discrete time steps** used to approximate the continuous probability flow. We have added these explanations to our revised paper.
>
> [1] Diffusion-Reward Adversarial Imitation Learning
>
> ---
> ## We sincerely hope your concerns are fully addressed.

---

> > ### Comment · Reviewer_XrqD · 2025-11-24
> >
> > Thank you for the detailed response. I still believe that the proposal to use FM as a discriminator——instead of a model that directly output the probability——is a good contribution. For the toy example regarding **multimodal training $\neq$ multimodal deployment**, could you provide some empirically results in a synthetic toy environment to evidence the discussion?

---

> > > ### Author Response · Authors · 2025-11-25
> > >
> > > Thank you  very much for the suggestion.
> > >
> > > To evidence this, we designed a controlled 1D Static Maze Experiment (fixed state $s=0$), and provide the results here for your convenience.
> > > * **Expert Data:** The expert provides actions from a bimodal distribution: $50\%$ clustered around $a=-1$ and $50\%$ around $a=1$. Note that the mean is $0$, but $a=0$ is actually a low-density (incorrect) action.
> > >
> > > **1. The FM Teacher’s Reward Landscape**
> > >
> > > First, we probed the learned FM-based reward function at three critical points. As shown in table below, the FM teacher successfully learns a multi-modal reward landscape. Crucially, it assigns low reward to the region between modes ($a=0$).
> > >
> > > | Evaluated Action ($a$) | **$a = -1$** (Left Mode) | **$a = 0$** (The "Average") | **$a = 1$** (Right Mode) |
> > > | :--- | :---: | :---: | :---: |
> > > | Ground Truth Density | High | Low | High |
> > > | FM-IRL Reward | 0.92 (High) | 0.12 (Low) | 0.86 (High) |
> > >
> > > **2. The Behavior of Student Policy**
> > >
> > > Next, we trained a standard uni-modal Gaussian policy ($\pi_\theta(a|s) = \mathcal{N}(\mu, \sigma)$) using this reward signal. Since the teacher says $a=0$ is "bad" (Reward 0.12), the student is forced to choose a side. The table below shows the final parameters of the converged student policy.
> > >
> > > | Student Parameter | Learned Value | Behavior Description |
> > > | :--- | :---: | :--- |
> > > | Mean ($\mu$) | -0.98 | Mode Seeking: Instead of hovering at $0$ (mode collapse/averaging), the policy drifted and locked onto the Left Mode ($\approx -1$). |
> > >
> > > **Conclusion:**
> > > This simple experiment confirms our hypothesis: the FM Teacher preserves the full multimodal structure (knowing that during **multi-modal training**, while the Student Policy uses this rich signal to find a single, consistent modal (**single-modal deployment**).
> > >
> > > ---
> > >
> > > We hope this toy example fully address your concerns.

---

> > > > ### Comment · Reviewer_XrqD · 2025-11-26
> > > >
> > > > Thank you for your efficient response. My understanding from the discussion of the revised paper is that in cases of imbalanced mode distributions (e.g., 40% left-first vs. 60% right-first), FM-IRL learns distribution-level distance metrics, while traditional methods may suffer from mode averaging failures. However, the current setting is balanced (50 samples per side) and lacks baselines. To convincingly support the discussion, this gap needs to be addressed, ideally by testing in an imbalanced setting and adding a baseline.
> > > >
> > > > I remain positive about this work and believe it opens up valuable research opportunities. However, I would encourage you to elaborate on this discussion with more convincing empirical evidence that directly supports the claims.

---

### Official Review · Reviewer_o29z · 2025-10-31

**Soundness:** 3
**Presentation:** 3
**Contribution:** 2
**Rating:** 4
**Confidence:** 4

**Summary:**

Flow Matching (FM) for RL is emerging as a strategy of Imitation Learning, which, however, defaults to offline learning, which lacks an exploration mechanism and is upper-bounded by expert demonstration performance. This paper proposed a method to use FM for reward shaping and regularization in online RL, which involves a teacher-student style learning. The teacher FM model shapes a reward for the agent model learning and action regularization during the online RL phase.

Empirical experiments on SOTA locomotion and navigation tasks showed that their method achieves higher generalization of learned policy and robustness to sub-optimal expert demonstrations in certain tasks.

**Strengths:**

+ This paper aims to tackle two fundamental challenges in online AIL: 1) where expert demonstrations are noisy or suboptimal ,and 2) traditional FM cannot adapt to an online setting
+ Preliminary work is clearly introduced to facilitate the introduction of the proposed method
+ The appendix offers an interesting theoretical discussion on why FM may offer advantages over conventional IRL reward models

**Weaknesses:**

Vague contribution scope:

* It looks to me that the main algorithmic novelty of this work is the integration of an FM model to replace the traditional IRL reward shaper, which feels incremental. Since the action regularization and the reward shaping methods can somewhat be traced back to prior work.   Especially, the discriminator training objective is identical to GAIL.
* The motivation for introducing FM into online IRL is relegated to the appendix rather than the main text. Although the argument there is compelling, the empirical evidence presented in the main paper does not strongly support these theoretical claims.

* Limited empirical improvement:
  - Reported performance improvements are marginal on several benchmarks (e.g., Hopper, Maze, Ant-goal)
  - For noisy initial and goal state settings, all experiments are evaluated only on a single task, Hand-rotate.
  - The experimental result in Table 1 does not support the claim that FM-IRL overcomes the limitation of suboptimal expert data, as they are approximately or often below the expected return of demonstration data.

**Questions:**

- Can a traditional  IRL discriminator be viewed as a special case of a  Flow Matching model, perhaps implicitly defining a probability flow between expert and policy distributions?
- What would be the algorithmic robustness in noisy settings for other locomotion/navigation tasks?

---

> ### Author Response · Authors · 2025-11-18
> **Author Response (1)**
>
> ## Thank you for the valuable suggestions. Here is our response to questions and concerns.
> ---
> **Q1.** The main algorithmic contribution looks vague and incremental, since the techniques used could be traced back to prior works.
>
> **A1.** We need to highlight that our key contribution is to introduce a **knowledge transfer architecture** that **decouples distribution modeling (teacher) from policy execution (student)**, enabling the Online FM-based learning without gradient instability (Figure 5 shows prior methods fail).
>
> Although reward shaping and regularization has been studied before, we design the reward shaping from a unique perspective-the issue of expert data multi-modality (please refer to our response to Q1 of review 1s8J for details). Meanwhile, we identified the potential shortcoming of such reward shaping and proposed a novel dual-FM regularization to address this issue, forming our well-grounded methodology **motivation-chain** rather than simple replace.
>
> ---
> **Q2.** The discriminator objective is identical to GAIL, making the contribution incremental.
>
> **A2.** This observation is critical, but it is exactly the **elegentness and advantage** of our method. GAIL is problematic when the expert data shows multi-modality property since it only captures point-level similarity, our method elegantly addressed this issue by simply optimize the structure of discriminator $D$. Future researcher can easily implement our method based on standard GAIL (only need to modify $D$ in GAIL objective following GAIL training pipeline).
>
> ---
> **Q3.** The motivation for introducing FM into online IRL is relegated to the appendix rather than the main text. There are no empirical evidence in main body of paper.
>
> **A3.** Actually, we have conducted a **case study** in our experiment session (l.423) to empirically prove the infeasibility of directly applying FM policy in online environment, highlighting the motivation of FM-IRL to utilize the expressiveness of FM but bypass the gradient computation instability of FM policy.
>
> ---
> **Q4.** There are only one environment in the generalization study.
>
> **A4.** Thanks for suggestion. We have added the experiment of another **Manipulation** task, Fetch-pick, in the main body of our revised paper (l.379, Figure 5), and **Navigation** and **Locomotion** tasks (see Table 4, l.1246) in Appendix F.3. The results show that FM-IRL exhibits much stronger generalization ability to states that are unseen in expert data, coinciding with our conclusion.
>
> We provide the results here for your convenience.
>
> - Manipulation: Fetch-pick (Avg Suc. Rate w.r.t noisy-level)
>
> | **Algo** | 1.00 | 1.25 | 1.50 | 1.75 | 2.00 | 2.25 |
> |-----------|------|------|------|------|------|------|
> | AIRL | 0.0 | 0.0 | 0.0 | 0.0 | 0.0 | 0.0 |
> | DP | 0.85 | 0.751 | 0.612 | 0.48 | 0.382 | 0.32 |
> | FP | 0.526 | 0.516 | 0.428 | 0.260 | 0.246 | 0.221 |
> | DRAIL | 0.70 | 0.658 | 0.24 | 0.0 | 0.0 | 0.0 |
> | GAIL | 0.252 | 0.115 | 0.0 | 0.0 | 0.0 | 0.0 |
> | VAIL | 0.859 | 0.915 | 0.14 | 0.015 | 0.0 | 0.0 |
> | WAIL | 0.0 | 0.0 | 0.0 | 0.0 | 0.0 | 0.0 |
> | **FM-IRL (ours)** | **1.0** | **0.92** | **0.90** | **0.88** | **0.87** | **0.85** |
>
> - Navigation: Maze (Avg Suc. Rate w.r.t expert coverage (%))
>
> | **Algo.** | **25** | **50** | **75** | **100** |
> |---|---|---|---|---|
> | AIRL | 0.8143 | 0.8528 | 0.9017 | 0.8729 |
> | FP | 0.5138 | 0.6247 | 0.7316 | 0.7584 |
> | DP | 0.5267 | 0.7419 | 0.8436 | 0.8917 |
> | DRAIL | 0.8059 | 0.8516 | 0.9128 | 0.8642 |
> | GAIL | 0.6534 | 0.6572 | 0.8127 | 0.8618 |
> | VAIL | 0.5926 | 0.7438 | 0.8015 | 0.9024 |
> | WAIL | 0.3184 | 0.1736 | 0.1652 | 0.5543 |
> | **FM-IRL (ours)** | **0.8273** | **0.9231** | **0.9426** | **0.9718** |
>
> - Locomotion: Walker2d (Avg Return w.r.t # demo transitions)
>
> | **Algo.** | **1k** | **1.5k** | **2k** | **2.5k** |
> |---|---|---|---|---|
> | AIRL | -7.23 | -2.51 | -5.94 | -4.62 |
> | FP | 1421.58 | 1827.93 | 2084.71 | 2228.49 |
> | DP | 1572.84 | 1896.25 | 2173.52 | 2468.37 |
> | DRAIL | 1458.76 | 1691.84 | 2947.39 | 3066.72 |
> | GAIL | 903.47 | 1158.65 | 1275.84 | 964.92 |
> | VAIL | 691.35 | 679.47 | 697.28 | 967.51 |
> | WAIL | 901.28 | 922.16 | 996.57 | 2457.85 |
> | **FM-IRL (ours)** | **3026.81** | **3635.42** | **3812.68** | **4368.93** |

---

> ### Author Response · Authors · 2025-11-19
> **Author Response (2)**
>
> **Q5.** The experiment results in some environments (e.g. Navigation tasks) show marginal performance improvement.
>
> **A5.** This observation exactly aligns with the motivation of FM-IRL. Please note that the advantage of our FM-enhanced discriminator lies in capturing multi-modal expert distributions during training. In navigation tasks, the objective is to move towards a target position. The success trajectory is often unimodal and is fully covered by expert data, obviating the need of excessive exploration and multi-modal training. Therefore, FM-IRL shows marginal improvement but  in such relatively "uni-modal tasks”, FM-IRL roughly **degraded to the strongest baseline, preserving its lower-bound performance**. In Manipulation tasks, the goal is not anymore “moving to a target place” but to try to manipulate an object, resulting in more complexity, flexibility and multi-modality in expert data. In such tasks, FM-IRL **significantly** outperforms baselines, validating our motivation. Meanwhile, our method introduces much-lower variance in tasks like Hopper, Fetch-pick and significant performance improvement in Walker2d, validating the effectiveness of regularization term to stabilize the training. We have added this discussion in Appendix B.1 (l.788).
>
> ---
> **Q6.** The result of robustness study does not support the claim that FM-IRL overcomes the limitation of suboptimal expert data, as they are often below the performance of expert data.
>
> **A6.** The robustness to sub-optimal data is **relative to baselines**, not experts. As an imitation learning approach, our policy's performance is inherently bounded by the expert data. However, when experts are sub-optimal, our method demonstrates superior robustness compared to DP and FP, underscoring the critical role of online exploration that motivates our work.
>
> ---
> **Q7.** Can a traditional IRL discriminator be viewed as a special case of a FM model?
>
> **A7.** No. Traditional IRL discriminator adopts MLP structure, taking inputs of (s,a) and directly outputs a scaler to measure the similarity. In MLP architecture, there are no probability flow. In our design, we use a optimal-trasnport FM to model the probability flow, but this is not directly utilized in measure the similarity. Instead, we adopts an implicit way: compute the average of probability loss in each layer of the discrete probability flow as a strong indicator of the distribution level similarity between (s,a) and expert data.
>
> ---
> ## We sincerely hope these responses fully address your concerns.

---

> > ### Comment · Reviewer_o29z · 2025-11-25
> > **Follow Up**
> >
> > Thanks to the authors for providing additional experimental results. While I am still conservative on the novelty and empirical performance of this work, I think the majority of the responses helped. Could the authors explain further why "The robustness to sub-optimal data is relative to baselines, not experts"? I couldn't find this original response convincing.
> >
> > ``A6. The robustness to sub-optimal data is relative to baselines, not experts. As an imitation learning approach, our policy's performance is inherently bounded by the expert data. However, when experts are sub-optimal, our method demonstrates superior robustness compared to DP and FP, underscoring the critical role of online exploration that motivates our work.``

---

> ### Author Response · Authors · 2025-11-26
> **Rebuttal Follow-up**
>
> ## Thank you for your active follow-up. We hereby address your concern in detail and step by step.
> ---
> - First, we need to clarify our setting: We adopt the **Inverse RL** setting, which belongs to the Imitation Learning setting, where we can only obtain expert data but not the environment's true reward. That is, we learn from expert demonstrations. Therefore, although our learning approach is online learning, our reward signal is still **completely provided by the dataset**.
>
> - In this situation, the performance of the learned policy is admittedly difficult to surpass that of the expert dataset (because the signal guiding its learning comes entirely from the expert dataset, without any other signal. It can be roughly considered that the performance upper bound of the learned policy is the expert's performance, restricted by the setting [1].
>
> - The robustness in our paper is **compared to baselines**, specifically Diffusion Policy and Flow-Matching Policy. You will find that when the expert data is sub-optimal, the performance of the baseline algorithms **directly collapses (far below the expert's)**, while the performance of our algorithm can **still approximate the expert's (and even surpass the expert in some cases, ID:2)**.
>
> - This experiment serves to verify our initial motivation: Supervised learning methods such as DP and FP lack online interaction, leading to their poor generalizability. When the expert data is sub-optimal, this limitation is **further amplified**, making it harder for them to generalize the expert's behavior when encountering unseen states, causing error accumulation and leading to performance collapse (far below the expert). Our method solves this problem, allowing the performance to approach the expert's upper bound when the expert data is sub-optimal.
>
> [1] Efficient reductions for imitation learning, 2011
>
> ---
> ## Thanks again for your active follow-up. We are willing to address your further concerns in our earliest effort, since the deadline of discussion is approaching.

---

### Official Review · Reviewer_1s8J · 2025-11-11

**Soundness:** 3
**Presentation:** 2
**Contribution:** 3
**Rating:** 2
**Confidence:** 3

**Summary:**

The paper proposes an adversarial imitation learning (AIL) method, termed FM-IRL, using flow-matching (FM) models. The main idea is to combine the benefits of adversarial imitation learning (which leads to better generalization and policy performance than behavior cloning) and FM-based policies (which are more expressive than MLP policies with a Gaussian action distribution). Since directly optimizing FM-based policies using AIL in an online manner is difficult, they first train an FM-based policy to clone the expert, and then train an MLP policy using a combination of (i) AIL reward derived from the FM-based policy, and (ii) regularization to stay close to the FM-based policy.

**Strengths:**

- The paper addresses an important research gap (i.e., how can we leverage flow-based policies for adversarial imitation learning)
- FM-IRL design choices (using a smart parameterization for the FM discriminator model, and a regularization term to stay close to the teacher FM policy) are clearly presented
- They show competitive performance against various AIL baselines, and report much better generalization to noise in the initial and goal states of the tasks

**Weaknesses:**

- Loss of multimodality in the trained policy: The main motivation of the paper is to combine the expressiveness of FM-based policies (i.e., their ability to represent multimodal policies) with AIL. However, since the student policy trained by FM-IRL is actually a unimodal MLP policy, it is unclear if the full potential of multi-modal policies is leveraged.
- Regularization in Eq. 11 may be mode-averaging: As a related point, I suspect the regularization term in equation 11 could lead to poor performance since it would encourage the unimodal MLP policy to spread its probability mass over multiple modes of the FM policy. It would be helpful to add an ablation study to test the benefit of the regularization.
- Unfair comparison to FM-based baselines: The comparison in Section 4.3 is great to have, but it might be unfair due to different rewards being used. I believe you are comparing baseline methods that use the sparse reward of the environment with FM-IRL, which uses a discriminator-based reward obtained using expert trajectories. Could you run the FM-A2C, FFM-PPO, and FPO baselines with the discriminator-based reward?
- Terminology (IRL vs. AIL): Even though the AIL problem is the same as IRL with a convex regularization, I would expect an IRL paper to have more empirical evaluation of the quality of the recovered reward, e.g., train a policy on the recovered reward and examine this policy's performance compared to the expert. In the absence of these experiments, I would recommend updating the name of the method to FM-AIL.

**Questions:**

None

---

> ### Author Response · Authors · 2025-11-18
>
> ## Thanks for the valuable insights and suggestions. Here are our responses.
> ---
> **Q1.** The policy during deployment is actually uni-modal MLP policy, then how does multi-modality leveraged?
>
> **A1.** We appreciate this insightful question, and we have added the discussion of this concern to the main body of our revised paper (see l. 469, Section 5). We also provide the justification here for your convenience.
>
> You are correct that the student policy is unimodal. However, our key insight is that **multimodal training ≠ multimodal deployment**.
> Consider an analogy: Suppose expert demonstrations contain two distinct navigation strategies (**modes**) through a maze—left-first (40% of expert data), right-first (60% of expert data). Traditional unimodal IRL methods (e.g., GAIL) use simple discriminators that cannot distinguish these modes' fine-grained differences, leading to coarse reward signals and potential **mode-averaging** failures (straight-first) [1].
> FM-IRL leverages multimodality differently: The FM teacher learns the complete multimodal structure of the expert distribution through flow matching. This rich understanding is encoded into the FM-enhanced discriminator via distribution-level distance metrics. When the student policy queries "Is action $a$ good at state $s$?", the discriminator provides rewards informed by knowledge of **all expert modes**—recognizing which mode(s) the current behavior resembles and how closely. Crucially, the student policy uses this multimodal-informed reward signal to discover **one single, optimal mode** for deployment through online exploration, rather than blindly averaging across modes. Actually, this is inspired by knowledge distillation [2], where a complex ensemble teacher's soft outputs, containing inter-class similarity structure, enable a simple student to achieve comparable performance.
>
> ---
> **Q2.** The ablation study of regularization term is required, and this term might cause mode-averaging and poor performance.
>
> **A2.** Thanks for suggestion. Actually, we have empirically justified the regularization term via ablation study in Appendix F.3 (l.1215). The results show that $\beta = 0$ (i.e. ablate regularization term) exhibits second-worst performance (convergence speed). From the rationale side, our regularization design specifically **avoids mode-averaging**. Please note that in practice, we sample only **one action** conditioning on a state from the FM generator. The student policy is penalized for deviating from this single sampled action, **not from all modes simultaneously**. Therefore, over many iterations, the student learns to commit to one consistent mode rather than averaging.
>
> ---
> **Q3.** The baselines in case study should use the same reward function as FM-IRL.
>
> **A3.**  Thanks for your valuable suggestion, and have conducted experiments comparing FM-IRL and three baseline in case study under the same reward setting (learned reward) in Appendix F.4 (l.1275). The results are similar to our original setting, and we provide the results here for your convenience:
>
> | Algorithm | Avg Success Rate |
> |:----------|:----------------:|
> | FM-A2C    | 0.0950 ± 0.04    |
> | FM-PPO    | 0.1250 ± 0.08    |
> | FPO       | 0.0533 ± 0.01    |
> | FM-IRL    | 0.8750 ± 0.03    |
>
> This is due to the **inherent instability of FM policy in online environment**, leading to poor performance regardless of the reward informativeness. This observation aligns with our claim in our case study that FM policies struggle in online environment, consolidating the design motivation of FM-IRL.
>
> ---
> **Q4.** The authors may change the name of the framework from "FM-IRL" to "FM-AIL".
>
> **A4.** Sorry for confusion. Indeed, our method employs adversarial training, which might suggest the name "FM-AIL.'' However, we deliberately chose "FM-IRL'' to reflect our adherence to the classical two-stage IRL paradigm: (1) learning a reward function $r_\theta(s,a)$ from expert demonstrations $\mathcal{D}\_E$, and (2) optimizing a policy $\pi_\phi$ using the learned reward via RL.  While GAIL [3] demonstrates that AIL is equivalent to IRL with entropy regularization, our method extends beyond standard AIL in a key aspect. Standard AIL derives rewards solely from discriminator outputs for occupancy measure matching. In contrast, FM-IRL incorporates a **dual mechanism**: (1) an FM-enhanced discriminator for reward modeling, and (2) explicit FM-based policy regularization. **This second component goes beyond the conventional AIL framework** and represents a distinct contribution that leverages FM's generative modeling capacity. We have added these discussion to our revised paper (l. 843).
>
> [1] InfoGAIL: Interpretable Imitation Learning from Visual Demonstrations, 2017
>
> [2] Distilling the knowledge in a neural network, 2015
>
> [3] Generative Adversarial Imitation Learning, 2016
>
> ---
> ## Thanks again for your comment. We are willing to address your further concern (if any) in our earliest effort.

---

### Author Response · Authors · 2025-12-02
**Summary of rebuttal (1)**

Dear ACs/PCs/SACs,

We understand that due to the recent incident, review scores have been reverted to their pre-discussion state. We sincerely appreciate the time and effort dedicated by all reviewers and by the previous AC to this paper. To facilitate your quick review of the current status, we provide a summary of the rebuttal updates and discussion outcomes, as well as the contribution of our paper.

**Core contribution**: Our paper introduces FM-IRL, a teacher–student framework that combines Flow Matching (FM) with Inverse Reinforcement Learning (IRL). The FM teacher captures the multimodal expert distribution and provides both reward modeling and regularization, while the student MLP policy explores online efficiently, avoiding gradient instability. This design enhances learning efficiency, generalization, and robustness, particularly when expert data is suboptimal.

Then, let us provide the most crucial information first: the ``final score distribution of our submission before the incident is 8,6,4,2`` with average score 5.



- - -
## Reviewer XrqD ``score:8 (soundness:3 + presentation: 3 + contribution 3), confidence: 3``
We are glad to see this reviewer seems to be satisfied with our work, and believes it opens up valuable research opportunities. Rather than expressing concerns, this reviewer mainly raised questions about the implementation difficulty and details, comparison fairness and hyperparameter settings, etc. We thank this reviewer for these valuable questions and provided detailed clarifications about these questions in our revised paper.

- - -
## Reviewer N3MV ``score:2->6 (soundness:2 + presentation: 3 + contribution 2), confidence: 3``
- This reviewer initially
- - criticized the paper as a simple combination of diffusion loss, distillation, and IRL, with weak motivation.
- - questioned whether rewards tied to offline data could generalize.

- We
- - have already acknowledged in our paper that our work is inspired by prior works but argue that inspired by doesn't mean mimicking. Following a rigorous logic and motivation chain, our architectural innovation is not a naïve combination but a teacher–student design that solves FM’s inability to update online.
- - We clarified that the reward model learns the expert’s objective rather than merely mimicking the dataset, drawing analogy to RL post-training in LLMs. FM provides directional reward signals that aid local OOD generalization, while regularization stabilizes exploration.

The reviewer further expressed the concerns about the impact of regularization weight $\beta$. We claimed that we already included a hyperparameter study of $\beta$ in our paper, and provided detailed explanation to this reviewer. After these clarifications, **we are happy to see the concerns of this reviewer are fully addressed, and this reviewer raised his score to 6.**


- - -
## Reviewer o29z ``score:4 (soundness:3 + presentation: 3 + contribution 2), confidence: 4``
- This reviewer felt
- - the experiment needs to be extended
- - FM as a reward shaper resembles prior work
- - the discriminator objective is identical to GAIL
- - motivation for FM was relegated to the appendix.
- - The performance of FM-IRL didn't exceed expert data, failing to demonstrate the robustness to sub-optimal data of FM-IRL.
- We emphasized that
- - We expanded experiments (Fetch-pick, Maze, Walker2d) in our revised paper to demonstrate stronger generalization and robustness of FM-IRL.
- - our key novelty lies in the teacher–student architecture, which decouples distribution modeling (teacher) from policy execution (student), enabling online FM learning without gradient instability.
- - The FM-enhanced discriminator addresses multimodality limitations of GAIL.
- - We highlighted a case study in main body of our paper showing FM policy instability in online settings, which forms the motivation of FM-IRL.
- - Our robustness is relative to baseline, not experts. Since we follow imitation learning setting, the expert performance is technically the upper-bound of any algorithm. However, our method shows stronger robustness compared to baselines.

Then, this reviewer expressed that the **questions are largely resolved, except one remaining concern** about the argument that "Our robustness is relative to baseline". We provided further clarifications to help this reviewer understand the imitation learning setting and our experiment expectation. **Unfortunately, the incident happens and we are unable to receive further feedback and the score re-consideration from this reviewer.**

---

### Author Response · Authors · 2025-12-02
**Summary of Rebuttal (2)**

---
## Reviewer 1s8J ``score:2 (soundness:3 + presentation: 2 + contribution 3), confidence: 3``
- The reviewer raised concerns that:
- - the unimodal student policy may lose FM’s multimodality
- - that the regularization term could cause mode averaging
- - that comparisons to FM baselines might be unfair (should use the same reward function)
- - that the method resembles Adversarial Imitation Learning (AIL) more than IRL, and encourage the authors to rename the framework.

- In response, we
- - clarified that multimodal training does not equal to multimodal deployment: the FM teacher captures multimodal structure, and its reward signal guides the unimodal student to commit to one optimal mode, similar to knowledge distillation.
- - We already showed through ablation (Appendix F.3) that regularization stabilizes training rather than causing averaging, and explained the rationale to this reviewer.
- - We also added experiments (Appendix F.4) with identical reward settings, where FM-IRL still significantly outperforms FM baselines.
- - Finally, we explained why we retain the name FM-IRL: the framework follows the classical IRL paradigm while extending beyond AIL by adding explicit FM-based policy regularization.

We hope these response could address the concern of this reviewer and the reviewer could re-consider the rating, but unfortunately **we didn't receive any feedback from this reviewer.**

- - -

# Closing words
On behalf of all the authors, I may again express our sincere gratitude for you to handle our submission. The incident is frustrating for all of us, but I understand it is harder time for you since the workload significantly increases. Regardless the final decision of this paper, hope these words could help us get through the hard time together.

Best regards,
Authors

---

### Meta-Review · Area_Chair_yiqa · 2025-12-04

**Summary:**

This paper proposes an adversarial imitation learning (AIL) method, termed FM-IRL, using flow-matching (FM) models. The main idea is to combine the benefits of adversarial imitation learning and FM-based policies. During learning, the proposed method first trains a BC FM policy and then trains an MLP policy using a combination of (i) AIL reward derived from the FM-based policy, and (ii) regularization to stay close to the FM-based policy.
The proposed method borrowed many ideas from DRAIL (NeurIPS 2024) and Flow Q-Learning (ICML 2025), e.g., using diffusion loss as reward, similar BC FM policy learning, and distillation with an MLP one-step policy to remedy learning difficulty. This work looks more like an extension of FQL to the AIL setting, rather than an original method.

The reviewers raised concerns primarily regarding the lack of novelty, potential mode-averaging behavior, marginal performance improvement, and insufficient evaluation. Although some of the reviewers' concerns have been addressed during rebuttal, the overall opinion of this paper is still mixed. Weighing all the pros and cons of the paper, I think the paper in its current shape is not ready to be accepted.

**Reviewer Concerns:**

Some concerns that I think are still not well-addressed:
- Lack of novelty and incremental contribution. Largely follows the GAIL recipe and borrows lots of items from FQL. (Reviewer o29z, N3MV, and AC)
- Marginal performance improvements and limited tasks (more task results are given during rebuttal, but still only 3 tasks). (Reviewer o29z, N3MV)
- Mode-averaging due to regularization in Eq. 11. (Reviewer 1s8J, XrqD)
- Unfair comparison to FM-based baselines. The authors only provide results of a single task, which might not be that convincing. (Reviewer 1s8J)

Some comments that I believe have been addressed or partly addressed during rebuttal, based on additional experimental results:
- Loss of multimodality in the trained policy due to distillation with an MLP student policy. (Reviewer 1s8J, XrqD)
- Concerns regarding ablations and implementation details from reviewers.

**Reviewer Scores:**

Reviewer N3MV mentioned that he/she increased the score from 2->6. Reviewer o29z mentioned that he is not satisfied with the authors' response.

I think 1s8J might increase his/her score of 2 a little bit, but might still lean on the negative side.

Overall, I think the final evaluations of the paper would still be mixed.

---

### Decision · Program_Chairs · 2026-01-26

Reject